# LOCALLY ADAPTIVE MULTI-OBJECTIVE LEARNING

## ABSTRACT

We consider the general problem of learning a predictor that satisfies multiple objectives of interest simultaneously. We work in an online setting where the data distribution can change arbitrarily over time. Here, multi-objective learning captures many common targets such as online calibration, regret, and multiaccuracy. In the online setting, common approaches to this problem that minimize the set of objectives over the *entire time horizon* can fail to adapt to distribution shifts. Previous work has tried to alleviate this problem by incorporating additional objectives that target local guarantees over contiguous subintervals. However, empirical evaluations of the performance of this proposal in practice are sparse. In this article, we consider an alternative procedure that achieves local adaptivity by replacing one part of the multi-objective learning method with an adaptive online algorithm. Empirical evaluations on datasets from energy forecasting and algorithmic fairness show that our proposed method improves upon existing proposals and achieves unbiased predictions over subgroups, while remaining robust under distribution shift.

## 1 INTRODUCTION

In an ever-changing world, real-time decision making necessitates coping with arbitrary distribution shifts and adversarial behavior. These shifts can arise from seasonality, change in data distribution induced by feedback loops or policy changes, and exogenous shocks such as pandemics or economic crises. Online learning is a powerful framework for analyzing sequential data that makes no assumptions on the data distribution.

Multi-objective learning is a generic framework that refers to any task in which a predictor must satisfy multiple objectives or criterion of interest simultaneously (Lee et al., 2022). This general framework has led to the development of online learning algorithms for numerous applications including multicalibration (Hebert-Johnson et al., 2018), multivalid conformal prediction (Gupta et al., 2022), and multi-group learning (Deng et al., 2024). Despite being a desirable and promising notion, methods from the online multi-objective learning literature have had little influence on the practice of machine learning.

We attribute this to two shortcomings. First, many of the algorithms proposed in the literature are not adaptive to abrupt changes in the data distribution: they learn a predictor that minimizes the objectives over the *entire time horizon*. In changing environments and in the presence of adversarial behavior, such algorithms will fail to cope with distribution shifts. Second, most prior work is purely theoretical with scant empirical evaluation. As a result, the practical aspects of multi-objective online algorithms have received limited consideration.

In this work, we aim to overcome the above shortcomings. We propose a locally adaptive multi-objective learning algorithm that outputs predictors which (approximately) satisfy a set of objectives over all local time intervals $I \subseteq [T]$. Previously, Lee et al. (2022) suggested a method that lends adaptivity to existing algorithms by including additional objectives for all contiguous subintervals. We present an alternative approach that directly modifies the multi-objective learning algorithm by replacing one part of the scheme with an adaptive online learning method. We provide a meta-algorithm that, given an adaptive online learner, minimizes the worst case multi-objective loss across time intervals. For concreteness, we instantiate it with the Fixed Share method (Herbster & Warmuth, 1998), which is guaranteed to provide adaptivity over all intervals of a fixed target width. Other possible instantiations of our approach that target alternative adaptive guarantees are discussed in Section 2.3.

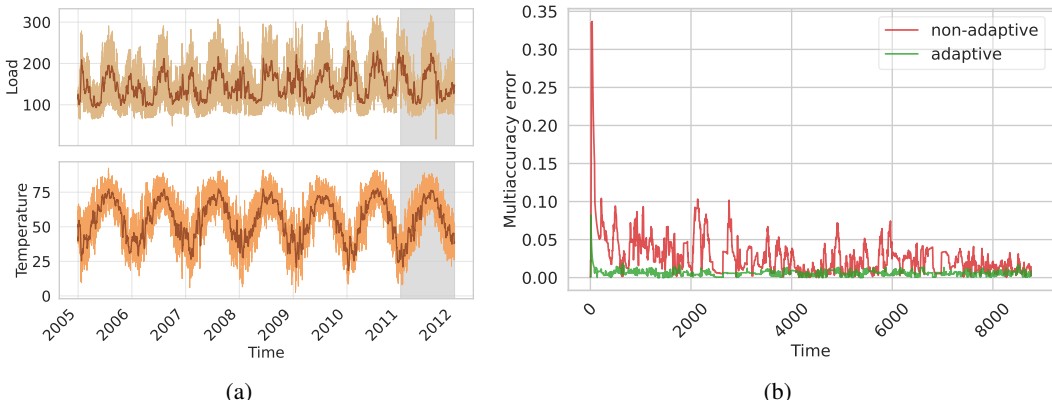

(a)                                                                    (b)

Figure 1: GEFCom14-L electric load forecasting dataset. On the left hand side are the time series for load and temperature. The dark brown curves indicate the weekly moving average. The shaded grey region shows the competition duration. On the right-hand side, we plot local multiaccuracy error.

To close the empirical gap in this literature, we provide extensive empirical evaluations comparing the performance of various adaptive methods in practice. This includes experiments on electricity demand forecasting and predicting recidivism over time in which our goal is to remove biases presenting in existing baseline predictors. Across all our empirical benchmarks we find that our proposed method consistently outperforms the previous proposals of Lee et al. (2022). We will release a codebase that implements our algorithm and all the baselines used in the paper.

As we discussed above, multi-objective learning can be used to address many common prediction tasks. As a case study, in this work, we focus on the multiaccuracy problem in which the goal is to learn predictors which are simultaneously unbiased under a set of covariate shifts of interest. We seek a small multiaccuracy error while preserving accuracy relative to a given sequence of baseline predictions. This is a problem of significant and broad interest across real-time decision-making and deployed machine learning systems. We show that our proposed algorithm has low multiaccuracy error over all intervals while the baselines have poor adaptivity. An alternative objective to multiaccuracy that is popular in the literature is multicalibration (Haghtalab et al., 2023a; Garg et al., 2024). Despite being a stronger condition, we show that in practice existing online multicalibration algorithms only achieve multiaccuracy at relatively slow rates. Adaptive extensions of the multicalibration algorithm yield improvements in local multiaccuracy error, however are unable to close the performance gap.

We note that although we focus on multiaccuracy in this paper, our general algorithm extends to other multi-objective learning problems including multi-group learning (Tosh & Hsu, 2022) and omniprediction (Gopalan et al., 2022). We discuss these extensions in Appendix A.

## 1.1 PEEK AT RESULTS

To demonstrate the significance of local adaptivity in practice, we consider the probabilistic electricity load forecasting track of the Global Energy Forecasting Competition 2014 (GEFCom2014) (Hong et al., 2016). The aim in the load forecasting track GEFCom2014-L is to forecast month-ahead quantiles of hourly loads for a US utility from January 1, 2011 to December 31, 2011 based on historical load and temperature data (Figure 1a).

We consider the binary task of predicting whether the electricity demand exceeds 150MW at hour $t$ and evaluate whether the predictions are multiaccurate with respect to discrete temperature groups $\{[0, 20), [20, 40), \ldots, [80, 100)\}$ (in °F). Informally, obtaining multiaccuracy with respect to temperature ensures our predictions are accurate at different times of day and across seasons. Figure 1b shows the multiaccuracy error of our proposed locally adaptive algorithm compared to a non-adaptive multiaccuracy algorithm, plotted as a weekly (168-hourly) rolling average. We can see that the multiacuracy error of the adaptive algorithm is close to zero across all time intervals, while the non-adaptive variant has high variance.

## 1.2 RELATED WORK

Our work is most closely related to the literature on multi-objective learning that encompasses numerous problems including multicalibration (Hebert-Johnson et al., 2018), multiaccuracy (Kim et al., 2019), multi-group learning (Tosh & Hsu, 2022), and omniprediction (Gopalan et al., 2022). Each of these multi-objective criteria have been studied in both the online and batch settings. Most closely related to our work, Kim et al. (2019) and Globus-Harris et al. (2023) give algorithms for obtaining multi-accurate and multi-calibrated (respectively) predictors in the batch setting that are guaranteed to have accuracy no worse than that of a given base predictor.

In the online advesarial setting, a number of works develop algorithms for obtaining multiaccuracy, multicalibration, and/or omniprediction globally over all time steps (Lee et al., 2022; Garg et al., 2024; Okoroafor et al., 2025; Haghtalab et al., 2023a; Noarov et al., 2025). Our work will in particular build on the algorithmic framework developed in Lee et al. (2022). This methodology has deep roots in the online learning literature and builds on ideas in blackwell approachability (Blackwell, 1956) and its connection to no-regret learning (Abernethy et al., 2011).

To obtain time-local guarantees we will draw on the literature on adaptive and strongly-adaptive regret (Herbster & Warmuth, 1998; Daniely et al., 2015; Jun et al., 2017; Haghtalab et al., 2023b). Our work will most closely rely upon the work of Gradu et al. (2023) to obtain multi-objective error bounds over any local time interval. In the context of multi-objective learning, local guarantees have been discussed previously in Lee et al. (2022). However, the literature contains no empirical evaluations of these methods. We provide experiments evaluating the algorithms of Lee et al. (2022) in Section 5 and find that our approach achieves significantly lower error rates in practice.

## 1.3 PRELIMINARIES

We use $\mathcal{X}$ to denote our feature space and $\mathcal{Y} = [a, b]$ to denote our label space, which we assume to be a bounded interval. Our goal is to learn a sequence of predictors $p_t \in \mathcal{Y}, t = 1, 2, \ldots, T$ that guarantee loss minimization simultaneously for every objective within a set $\mathcal{L}$ over time. Each objective, or criterion, is a function $\ell : \mathcal{Y} \times \mathcal{X} \times \mathcal{Y} \to [-1, 1]$ that takes as input a predictor $p_t$, features $x_t \in \mathcal{X}$, and label $y_t \in \mathcal{Y}$ and returns a value in $[-1, 1]$. We will use $[T]$ to denote the set $\{1, 2, \ldots, T\}$.

The objectives we consider can be quite general and we will give some examples of specific choices shortly. Broadly, our only restriction is that the objectives should be consistent with one another in the sense that for any distribution on $y_t$ there is a single optimal predictor $p_t$ that minimizes all the objectives simultaneously. Formally, we assume the following.

**Assumption 1.** *For any $x \in \mathcal{X}$ and distribution $P_Y$ on $\mathcal{Y}$ there exists $p^* \in \mathcal{Y}$ such that for all $\ell \in \mathcal{L}$,*

$$p^* \in \operatorname*{argmin}_{p \in \mathcal{Y}} \mathbb{E}_{Y \sim P_Y}[\ell(p, x, Y)].$$

*Moreover, for all $\ell \in \mathcal{L}$, $p^*$ guarantees the loss bound*

$$\mathbb{E}_{Y \sim P_Y}[\ell(p^*, x, Y)] \leq 0. \tag{1}$$

The assumption that $p^*$ produces a negative objective value is not strictly necessary and previous work in multiobjective learning has considered slightly more general settings (Lee et al., 2022). We have chosen to add this condition because it simplifies the notation and is satisfied by many common problems of interest. For instance, as we will discuss in the sections that follow, multiaccuracy, multicalibration, omniprediction, and multi-group learning can all be formulated in a way that meets this condition.

Using this assumption, our goal will be to learn a sequence of predictors $p_t$ that (approximately) matches the optimal bound (1). We study this problem in the following online, adversarial setting.

**Definition 1** (Online multi-objective learning). *For a set of objectives $\mathcal{L} = \{\ell : \mathcal{Y} \times \mathcal{X} \times \mathcal{Y} \to [-1, 1]\}$ and sequence of data points $x_t, y_t, t \in [T]$, the goal of online multi-objective is to learn a sequence of predictors $p_t$ such that*

$$\max_{\ell \in \mathcal{L}} \frac{1}{T} \sum_{t=1}^{T} \ell(p_t(x_t), x_t, y_t) \lessapprox 0,$$

where $(x_t, y_t)$ can be generated advesarially dependent on the entire history of data and predictions up to time $t$.

As an example, we now define two instantiations of multi-objective problems that are commonly studied in the literature and which we will focus on—multiaccuracy and multicalibration. The offline version of multiaccuracy was introduced in Kim et al. (2019). We parameterize the multiaccuracy criterion by a function class $\mathcal{F}$ and the goal is to be unbiased for all $f \in \mathcal{F}$, i.e., there is no systematic correlation between the prediction residuals and any $f \in \mathcal{F}$.

**Definition 2** (Online multiaccuracy). Let $\mathcal{F} = \{f : \mathcal{X} \to [0, 1]\}$ be a class of functions on $\mathcal{X}$. In online multiaccuracy, we instantiate $\ell_{\mathrm{MA}_{f,\sigma}}(p_t(x_t), x_t, y_t) = \sigma f(x_t) \cdot (y_t - p_t(x_t))$ for every sign $\sigma = \{\pm\}$ and $f \in \mathcal{F}$ and define the multiaccuracy error $\ell_{\mathrm{MA}}$ in the sup-norm as

$$\ell_{\mathrm{MA}}(p_t(x_t), x_t, y_t) = \sup_{f \in \mathcal{F}, \sigma \in \{\pm\}} \frac{1}{T} \sum_{t=1}^{T} \sigma f(x_t) \cdot (y_t - p_t(x_t)). \tag{2}$$

Another popular online prediction target is calibration. In a binary classification task, calibration asks that among instances with predicted probability $p$, a fraction $p$ of them are observed to be truly labeled as 1. To implement this in practice, we discretize the label interval $[0, 1]$ into $m$ bins $V_m := \{[0, 1/m], [1/m, 2/m), \ldots, [(m-1)/m, 1]\}$ and define a representative value for each bin as the midpoint $v_j = \frac{2j-1}{2m}$ for $j = 1, \ldots, m-1$. Calibration asks that for each bin $v \in V_m$, the empirical proportion with $y_t = 1$ among points $p_t \in v$ is close to $v_j$. Multicalibration is a stronger notion that requires a predictor to be calibrated under all reweightings $f \in \mathcal{F}$. Standard calibration is then the special case where $\mathcal{F} = \{x \mapsto 1\}$ is a singleton function class containing only the identity. The offline version of multicalibration was introduced in Hebert-Johnson et al. (2018).

**Definition 3** (Online multicalibration). Fix a set of functions $\mathcal{F}$ and $m \geq 1$. In online multicalibration we instantiate $\ell_{\mathrm{MC}_{f,\sigma,v}}(p_t(x_t), x_t, y_t) = \sigma f(x_t) \cdot \mathbb{1}\{p_t(x_t) \in v\} \cdot (y_t - v_j)$ for every sign $\sigma = \{\pm\}$, $f \in \mathcal{F}$, and $v \in V_m$ and define the multicalibration error $\ell_{\mathrm{MC}}$ in the sup-norm as

$$\ell_{\mathrm{MC}}(p_t(x_t), x_t, y_t) = \sup_{f \in \mathcal{F}, \sigma \in \{\pm\}, v \in V_m} \frac{1}{T} \sum_{t=1}^{T} \sigma f(x_t) \cdot \mathbb{1}\{p_t(x_t) \in v\} \cdot (y_t - v_j). \tag{3}$$

A direct calculation shows that the online multicalibration error always upperbounds the multiaccuracy error; specifically, $\ell_{\mathrm{MA}} \leq m \cdot \ell_{\mathrm{MC}}$.

In this work, we investigate multi-objective learning algorithms that achieves small multiaccuracy error while preserving accuracy relative to a base predictor sequence $\tilde{p}_t, t \in [T]$. We define the latter accuracy objective as regret

$$\ell_{\mathrm{reg}}(p_t(x_t), x_t, y_t) := \frac{1}{T} \sum_{t=1}^{T} \ell_{\mathrm{acc}}(p_t(x_t), y_t) - \ell_{\mathrm{acc}}(\tilde{p}_t(x_t), y_t), \tag{4}$$

where $\ell_{\mathrm{acc}} \geq 0$ is any proper loss for the mean, i.e., any loss such that $\mathbb{E}_{y \sim P}[y] \in \operatorname{argmin}_p \mathbb{E}_{y \sim P}[\ell(p, y)]$ for all distributions $P$ on $\mathcal{Y}$. A common example that we will work with in our experiments is the squared error/Brier score $\ell_{\mathrm{acc}}(p, y) = (y - p)^2$.

## 2 METHODS

### 2.1 ONLINE MULTI-OBJECTIVE LEARNING

The online multi-objective learning problem is a sequential prediction task over $T$ rounds. A common approach introduced in Lee et al. (2022) is to consider a two-player game between a learner, who observes $x_t \in \mathcal{X}$ and chooses a predictor $p_t$, and an adversary who maintains a distribution $q^{(t)} \in \Delta(\mathcal{L})$, where we use the notation $\Delta(S)$ to denote the set of probability distributions over the set $S$. At each time step, the learner observes the adversary's current mixture and the covariates $x_t$ and chooses its (randomized) prediction as

$$p_t(x_t) \sim P_t(x_t), \quad \text{where} \quad P_t(x_t) = \operatorname*{argmin}_{P \in \Delta(\mathcal{Y})} \max_{y \in \mathcal{Y}} \mathbb{E}_{p \sim P} \left[ \sum_{\ell} q_{\ell}^{(t)} \ell(p, x_t, y_t) \right].$$

This choice is designed to guarantee that the learner obtains the best possible performance under the adversarial value of $y_t$ with respect to the mixture loss specified by $q^{(t)}$. As an aside, we note that although generic multiobjective learning problems require randomized predictors, our methods will often produce deterministic values. This is due to the fact that for many of the problems we are interested in (e.g., multiaccuracy, low predictive accuracy) the objectives are convex and thus the minmax program above admits a deterministic solution.

After the learner makes its selection, the true value of $y_t$ is revealed and the adversary updates its mixture distribution. In the original work of Lee et al. (2022), the adversary sets its weights using the hedge updates

$$q_\ell^{(t+1)} \propto q_\ell^{(t)} \exp(\eta \ell(p_t(x_t), x_t, y_t)),$$

for some $\eta = \Theta(\sqrt{\log(|\mathcal{L}|)/T})$. This is designed to ensure that the mixture distribution with respect to $q^{(t)}$ is a good proxy for the maximum multiobjective error. More formally, this choice of weights has the well-known error bound (e.g. Theorem 1.5 of Hazan (2016)),

$$\max_{\ell \in \mathcal{L}} \sum_{t=1}^{T} \ell(p_t(x_t), x_t, y_t) \le \sum_{\ell \in \mathcal{L}} \sum_{t=1}^{T} q_\ell^{(t)} \ell(p_t(x_t), x_t, y_t) + O(\sqrt{T \log(|\mathcal{L}|)}).$$

By combining this bound with the choice of $p_t$ we obtain the following multiobjective error bound.

**Theorem 1** (Theorem 2.1 in Lee et al. (2022)). *Under Assumption 1, Algorithm 1 with hedge as the method for learning $q^{(t)}$ obtains the multiobjective learning bound*

$$\max_{\ell \in \mathcal{L}} \sum_{t=1}^{T} \ell(p_t(x_t), x_t, y_t) \le O(\sqrt{T \log(|\mathcal{L}|)})$$

## 2.2 LOCALLY ADAPTIVE MULTI-OBJECTIVE LEARNING

The result of Theorem 1 ceases to be useful when environments are changing and the data distribution shifts arbitrarily over time. As a simple example, fix the singleton function class $\mathcal{F}_{\mathrm{MA}} = \{x \mapsto 1\}$ and consider targeting just the multiaccuracy error (i.e., set $\mathcal{L} = \{\ell_{\mathrm{MA}_{f,\sigma}} : f \in \mathcal{F}_{\mathrm{MA}}, \sigma \in \{\pm\}\}$). Let the labels be given as $y_t = 1$ for the first $T/2$ rounds and $y_t = 0$ for the last $T/2$ rounds. Here, the constant predictor $p_t = 1/2$ minimizes the multiaccuracy error in (2). Nevertheless, this predictor performs poorly in the individual intervals $1 \le t \le T/2$ and $t > T/2$ compared to the optimal predictor that switches from $p_t = 1$ to $p_t = 0$ after $t = T/2$.

To account for distribution shifts in changing environments, we will now modify the method of Lee et al. (2022) by replacing the hedge algorithm with a locally adaptive method. Informally, this will allow us to bound the worst case multi-objective loss over intervals given by

$$\sup_{I=[r,s]} \left[ \max_{\ell \in \mathcal{L}} \sum_{t=r}^{s} \ell(p_t(x_t), x_t, y_t) \right], \tag{5}$$

where the supremum is over some appropriate set of intervals $I$ that we will specify shorty. Algorithm 1 gives our generic method. Here, WL denotes any procedure for learning the weights $q^{(t)}$. Some example are discussed in the next section.

---

**Algorithm 1** Locally adaptive multi-objective learning

---

**Input:** Set of objectives $\mathcal{L}$, learning method WL
**Input:** Sequence of samples $\{(x_1, y_1), \ldots, (x_T, y_T)\}$
1: $q_\ell^{(1)} = \frac{1}{|\mathcal{L}|}, \quad \forall \ell \in \mathcal{L}.$
2: **for** each $t \in [T]$ **do**
3: $\quad P_t(x_t) = \operatorname*{argmin}_{P \in \Delta(\mathcal{Y})} \max_{y \in \mathcal{Y}} \mathbb{E}_{p \sim P} \left[ \sum_{\ell \in \mathcal{L}} q_\ell^{(t)} \ell(p, x_t, y_t) \right]$
4: $\quad$ Output $p_t(x_t) \sim P_t(x_t)$
5: $\quad q_\ell^{(t+1)} = \mathtt{WL}(\{q^{(s)}\}_{s \le t}, \{\mathbb{E}_{p \sim P_t(x_t)}[\ell(p, x_t, y_t)]\}_{\ell \in \mathcal{L}})$

---

---

**Algorithm 2** Fixed-Share weight update

---

**Input:** Weights at current timestep $q^{(t)}$; hyperparameters $\eta, \gamma$.
**Input:** Losses for current timestep $\{\mathbb{E}_{p \sim P_t(x_t)}[\ell(p, x_t, y_t)]\}$
  1: **for** each $t \in [T]$ **do**

  2:    $\tilde{q}_\ell^{(t+1)} = \dfrac{q_\ell^{(t)} \exp\left(\eta \cdot \mathbb{E}_{p \sim P_t(x_t)}[\ell(p, x_t, y_t)]\right)}{\sum_{\ell' \in \mathcal{L}} q_{\ell'}^{(t)} \exp\left(\eta \cdot \mathbb{E}_{p \sim P_t(x_t)}[\ell'(p, x_t, y_t)]\right)}$, for all $\ell \in \mathcal{L}$

  3:    $q_\ell^{(t+1)} = (1 - \gamma)\, \tilde{q}_\ell^{(t+1)} + \dfrac{\gamma}{|\mathcal{L}|}$
**Output:** Weights for the next time step $q^{(t+1)}$

---

## 2.3 A CLOSER LOOK AT ADAPTIVE ALGORITHMS IN LITERATURE

Adaptive online learning algorithms guarantee low loss with respect to an optimal sequence of predictors over all contiguous time intervals. Formally, this is defined as the *adaptive regret* of an algorithm. Lee et al. (2022) propose an adaptive extension of the multi-objective learning algorithm by including additional objectives for all subintervals. Formally, given an initial set of objectives $\mathcal{L}$ they consider the augmented collection $\mathcal{L}_{\text{adapt.}} = \{\ell(p_t(x_t), x_t, y_t)\mathbb{1}\{t \in I\}, \ell \in \mathcal{L}, I = [r, s] \subseteq [T]\}$ and show that using these objectives with the alongside the algorithm described in Section 2.1 guarantees the local bound

$$\sup_{I=[r,s]\subseteq[T]} \sum_{t \in [I]} \mathbb{E}_{p \sim P_t(x_t)}\ell(p, x_t, y_t) \leq O(\sqrt{T(\log(|\mathcal{L}|) + 2\log T)},$$

where the supremum is over all contiguous intervals $I \subseteq [T]$. In our work, we propose using a locally adaptive procedure WL to learn the weights $q^{(t)}$. As a concrete instantiation, we will perform empirical experiments on the Fixed Share method introduced in Herbster & Warmuth (1998) that modifies the hedge update by adding an exploration term that stops any of the the weights from collapsing to zero. A formal statement of this procedure is given in Algorithm 2. As we will discuss in the next section, Fixed Share is guaranteed to perform as well as the best expert *locally* on any interval of a fixed width. There are many possible alternative methods that one could implement in the place of Fixed Share. For instance, one may consider the *strongly adaptive* learning procedure of Daniely et al. (2015) and Jun et al. (2017) that guarantee a stronger notion of adaptive regret with dependency over the interval width $|I|$ for all intervals $I \subseteq [T]$. We have chosen to focus on Fixed Share due to its strong empirical performance.

## 3 THEORY

We will now state a theoretical guarantee for Algorithm 1. For concreteness, we will focus on the case where the adversary learns the weights $q^{(t)}$ using the Fixed Share method given in Algorithm 2. Similar results for other adaptive learning methods can be obtained in an identical fashion by replacing the regret bound for Fixed Share (Lemma 1 below) with the associated bound for that method.

The theory has two parts: guarantees for the adversary's distribution $q$ and guarantee of the learner's response. From here on, we use the shorthand $\ell^{(t)} := \mathbb{E}_{p \sim P_t(x_t)}[\ell(p, x_t, y_t)]$ to denote the expected loss of our randomized predictor at time step $t$ and denote the $|\mathcal{L}|$-dimensional vector of losses as $\ell_{\mathcal{L}}^{(t)} = (\ell^{(t)})_{\ell \in \mathcal{L}}$. All proofs are deferred to Appendix B.

We first show that the maximum objective value over any time interval $I$ is upper bounded by the average value of the individual objectives taken with respect to the weights $q^{(t)}$.

**Lemma 1.** *Consider Algorithm 1 with weights learned using Algorithm 2. Assume that $\gamma \leq 1/2$. Then, for any interval $I = [r, s] \subseteq [T]$,*

$$\sum_{t=r}^{s} q^{(t)\top}\ell_{\mathcal{L}}^{(t)} \geq \max_{\ell \in \mathcal{L}} \sum_{t=r}^{s} \ell^{(t)} - \eta\left(\sum_{t=r}^{s} q^{(t)\top}(\ell_{\mathcal{L}}^{(t)})^2\right) - \frac{1}{\eta}\left(\log\left(\frac{|\mathcal{L}|}{\gamma}\right) + |I|2\gamma\right). \tag{6}$$

Next, we show that the average value of the objectives is non-positive over any interval $I$. This lemma follows from the minimax-optimal strategy of the learner and has been shown to hold previously in Lee et al. (2022).

**Lemma 2.** *Suppose the objectives satisfy Assumption 1. Then, for any interval $I = [r, s] \subseteq [T]$,*

$$\sum_{t=r}^{s} q^{(t)\top} \ell_{\mathcal{L}}^{(t)} \leq 0.$$

We combine the previous two lemmas to get our main result.

**Theorem 2.** *Assume that $\gamma \leq 1/2$ and the objectives satisfy Assumption 1. Then, for any interval $I = [r, s] \subseteq [T]$,*

$$\max_{\ell \in \mathcal{L}} \frac{1}{|I|} \sum_{t=r}^{s} \ell^{(t)} \leq \frac{\eta}{|I|} \left( \sum_{t=r}^{s} q^{(t)\top} (\ell_{\mathcal{L}}^{(t)})^2 \right) + \frac{1}{\eta|I|} \left( \log \left( \frac{|\mathcal{L}|}{\gamma} \right) + |I| 2\gamma \right). \tag{7}$$

The guarantee of Theorem 2 depends on the values of the fixed share hyperparameters $\gamma, \eta$. To set the best upperbound for a given interval $I$, we would ideally substitute the optimal values $\gamma = \dfrac{1}{2|I|}$ and $\eta = \sqrt{\dfrac{\log(|\mathcal{L}| \cdot 2|I|) + 1}{\sum_{t=r}^{s} q^{(t)\top} (\ell_{\mathcal{L}}^{(t)})^2}}$ in (7) and obtain

$$\max_{\ell \in \mathcal{L}} \frac{1}{|I|} \sum_{t=r}^{s} \ell^{(t)} \leq \frac{2}{|I|} \sqrt{\left( \log(|\mathcal{L}| \cdot 2|I|) + 1 \right)} \cdot \sqrt{\sum_{t=r}^{s} q^{(t)\top} (\ell_{\mathcal{L}}^{(t)})^2} = O\left( \sqrt{\frac{\log(|\mathcal{L}| \cdot |I|)}{|I|}} \right) \tag{8}$$

In practice, we can only use one setting of these parameters and cannot specialize $\gamma$ and $\eta$ to a specific interval. To mimic these optimal choices, we let the user pick a fixed target interval width $|I| = \tau$, noting that a smaller choice of $\tau$ gives stronger locally adaptive guarantees at the cost of a looser upper bound. Since the optimal value for $\eta$ used above depends on the expected squared objective $\sum_{t=r}^{s} q^{(t)\top} (\ell_{\mathcal{L}}^{(t)})^2$ which is unknown in practice, we follow Gibbs & Candès (2024) in selecting an adaptive value of $\eta$ that updates online as

$$\eta = \eta_t := \sqrt{\frac{\log(|\mathcal{L}| \cdot 2\tau) + 1}{\sum_{s=t-\tau+1}^{t} q^{(s)\top} (\ell_{\mathcal{L}}^{(s)})^2}}. \tag{9}$$

This choice lets the algorithm adaptively track changes in the moving average of the expected squared objective over the most recent $\tau$ time steps.

## 4 CASE STUDY: MULTIACCURACY

As a case study, we focus on the multiaccuracy problem in this work. Our goal is to learn predictors that have small multiaccuracy error (2) while guaranteeing the prediction error (4) is low relative to a given sequence of baseline predictions. We fix a function class $\mathcal{F}_{\text{MA}} \subseteq \{f : \mathcal{X} \to [0, 1]\}$ that we desire multiaccuracy with respect to and define $\mathcal{L} := \{\ell_{\text{MA}_{f,\sigma}} : f \in \mathcal{F}_{\text{MA}}, \sigma \in \{\pm\}\} \cup \{\ell_{\text{reg}}\}$ including the regret objective. We use the shorthands $\ell_{\text{MA}_{f,\sigma}}^{(t)} := \sigma f(x_t)(y_t - p_t(x_t))$ and $\ell_{\text{reg}}^{(t)} := \ell_{\text{acc}}(p_t(x_t), y_t) - \ell_{\text{acc}}(\tilde{p}_t(x_t), y_t)$ to denote the realized losses, noting that since these objectives are convex we may assume without loss of generality that the predictor $p_t(x_t)$ is deterministic. We provide an algorithm for locally-adaptive multiaccurate mean estimation in Algorithm 4 in the appendix and its guarantee in Corollary 1.

**Corollary 1.** *Assume that $\gamma \leq 1/2$. Then, for any interval $I = [r, s] \subseteq [T]$,*

$$\max \left\{ \max_{f,\sigma} \frac{1}{|I|} \sum_{t=r}^{s} \ell_{\text{MA}_{f,\sigma}}^{(t)}, \frac{1}{|I|} \sum_{t=r}^{s} \ell_{\text{reg}}^{(t)} \right\} = O\left( \sqrt{\frac{\log((2|\mathcal{F}_{\text{MA}}| + 1) \cdot |I|)}{|I|}} \right).$$

Next, we discuss the importance of including the regret objective in multiaccuracy problems.

### 4.1 SIGNIFICANCE OF THE REGRET OBJECTIVE

In our applications, we will start with a base forecaster, $\tilde{p}_t$ that was constructed in advance for that application. Our goal will be to improve $\tilde{p}_t$ to be multiaccurate. While doing this, it is important that we do not degrade the accuracy of $\tilde{p}_t$, thereby rendering its predictions less useful. Our algorithm achieves small multiaccuracy error while preserving the accuracy relative to a base predictor by including an additional regret objective (4).

In general, even in the absence of a base predictor, it is not advisable to solely target multiaccuracy. Indeed, if we exclude the regret objective in Algorithm 4 one can show that the best response of the learner yields the predictor: $p_t = b\mathbb{1}\big\{\sum_{f,\sigma} q^{(t)}_{\mathrm{MA}_{f,\sigma}} \sigma f(x_t) > 0\big\} + a\mathbb{1}\big\{\sum_{f,\sigma} q^{(t)}_{\mathrm{MA}_{f,\sigma}} \sigma f(x_t) \leq 0\big\}$. This solution has a pathological behavior where the predictor will only take the extreme values $a$ or $b$ at every step. This makes the predictions less useful and interpretable for real-time decision-making in an online setting. Our regret objective recovers the predictor from this problem by enforcing solutions that do not lie in the extremes. In practical settings where $\tilde{p}_t$ is not available in advance, we recommend combining our procedure with a standard online learning algorithm (e.g., online gradient or mirror descent) that provides an appropriate baseline (see e.g. Algorithm 3 in the appendix).

## 5 EXPERIMENTS

### 5.1 DATASETS

**GEFCom2014 electric load forecasting.** Global Energy Forecasting Competition 2014 (GEFCom2014) (Hong et al., 2016) was a probabilistic energy forecasting competition conducted with four tracks on load, price, wind and solar forecasting. In this work, we study the electricity demand forecasting track GEFCom2014-L. In Section 1.1, we shared details regarding the task and Figure 1a displays the load and temperature trends over time. We set function class $\mathcal{F}$ to be the temperature groups $\{[0, 20), [20, 40), \ldots, [80, 100)\}$ (in °F). We consider a binary load prediction task for our empirical evaluation, in which the goal is to estimate the probability that electricity demand exceeds 150 MW during hour $t$. We construct our baseline predictions $\tilde{p}_t$ by linearly interpolating the quantiles forecasts of Ziel & Liu (2016), whose method outperform top entries of the competition. See Appendix D.1 for details of the linear interpolation procedure.

**COMPAS dataset.** Larson et al. (2016) analyzed the COMPAS tool used to predict recidivism for criminal defendants in Broward County, Florida and found that certain groups of defendants are more likely to be incorrectly judged as high risk of recidivism. In Figure 16, we plot the true recidivism rate over time for different racial groups. We consider the recidivism prediction task and evaluate the local multiaccuracy of predictors with respect to the African-American, Caucasian, and Hispanic subgroups. We use the COMPAS risk scores provided in the dataset as our baseline predictions. Following the analysis of Barenstein (2019) who point the data processing error in the two-year sample cutoff rule for recidivists, we drop the data points beyond this two year cutoff.

### 5.2 BASELINES

We consider baselines that differ in their adaptivity and the set of objectives in $\mathcal{L}$. **MA+reg** denotes the algorithm with the multiaccuracy and regret objectives $\mathcal{L} := \{\ell_{\mathrm{MA}_{f,\sigma}} : f \in \mathcal{F}_{\mathrm{MA}}, \sigma \in \{\pm\}\} \cup \{\ell_{\mathrm{reg}}\}$. We explain the baselines below:

**Baseline predictors $\tilde{p}_t$.** These are the predictions that were constructed in advance for the application and are our input to Algorithm 4.

**Multiaccuracy (MA)** with $\mathcal{L} := \{\ell_{\mathrm{MA}_{f,\sigma}} : f \in \mathcal{F}_{\mathrm{MA}}, \sigma \in \{\pm\}\}$: This is a specific case of Algorithm 4 where the set $\mathcal{L}$ does not include the regret objective.

**Multicalibration (MC).** We implement the online multicalibration algorithm from Lee et al. (2022). This is a competitive algorithm as multicalibration is a stronger condition than multiaccuracy. Lee et al. (2022) show that their algorithm can guarantee that predictions satisfy an accuracy objective (specifically, low squared error) on subgroups in addition to multicalibration. Hence, we consider

$\ell_{\text{acc}}$ as the squared error in our regret objective. We take the number of bins as $m = 10$ as it is a reasonable target for multicalibration. Lower values will give better multiaccuracy results at the cost of a much weaker multicalibration guarantee. We evaluate for varying $m$ in Appendix D.2.

We consider three variants for the algorithms: **non-adaptive, locally adaptive**, and **adaptive**. The non-adaptive variant corresponds to using hedge to learn the weights in Algorithm 1; the locally adaptive variant corresponds to using the Fixed Share update as stated in Algorithm 2; and the adaptive variant corresponds to using hedge with additional objectives for all subintervals. Specifically, the adaptive method augments the objectives as $\mathcal{L}_{\text{adapt.}} = \{\ell(p_t(x_t), x_t, y_t)\mathbb{1}\{t \in I\}, \ell \in \mathcal{L}, I = [r, s] \subseteq [T]\}$.

### 5.3 LOCAL MULTIACCURACY AND REGRET EVALUATION

In this section, we evaluate the local multiaccuracy error $\ell_{\text{MA}}$ and regret $\ell_{\text{reg}}$ incurred by the algorithms we defined above.

First, we look at the results on GEFCom2014-L dataset (Figure 2). We take the interval width $\tau = 336$ hours (2 weeks) for this set of experiments. We show results with varying $\tau$ in Appendix F.3. We compute empirical local multiaccuracy and regret rates over this moving 2 week-window. It can be seen that the constructed baseline predictor $\tilde{p}_t$ has high local multiaccuracy error and all algorithms improve over this baseline. Overall, the locally adaptive algorithms (MA and MA+reg) have close to zero multiaccuracy over all local intervals. On the other hand, the non-adaptive algorithms have high local variability. It is interesting to note that both the non-adaptive and the locally adaptive variants of the multicalibration algorithm (MC) have significantly slower multiaccuracy rates in practice.

Next, we turn to study the empirical local regret of these algorithms. As expected, the MA baseline has non-zero regret in some local intervals and we lose accuracy with respect to the predictor $\tilde{p}_t$ in the absence of the regret objective. MA+reg consistently preserves or improves accuracy over $\tilde{p}_t$. As promised by the multicalibration+multicalibeating algorithm in Lee et al. (2022), we observe that MC generally has negative regret, although with poorer adaptivity compared to MA+reg.

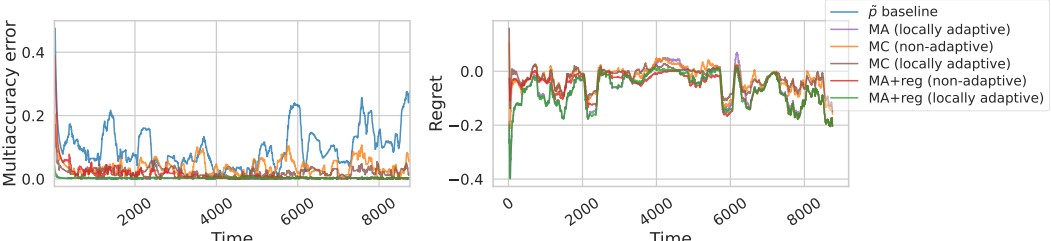

Figure 2: Local multiaccuracy error and regret on GEFCom2014-L dataset. We skip the first ten time steps when plotting the multiaccuracy error and regret for improved readability.

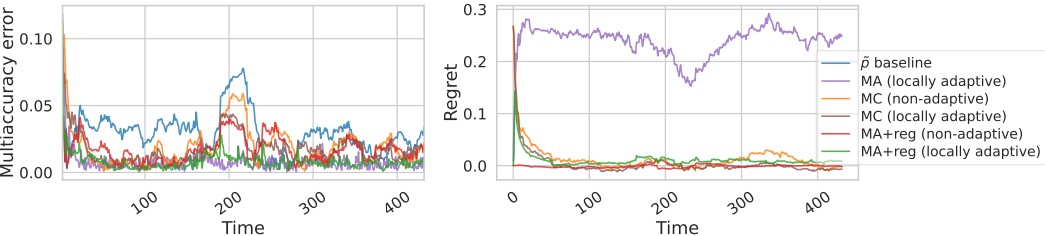

Figure 3: Local multiaccuracy error and regret on COMPAS dataset. We skip the first two time steps when plotting the multiaccuracy error for improved readability.

Now, we examine our results on the COMPAS dataset (Figure 3). Here, we fix $\tau = 50$ days. Our findings from above are seen to generalize here. $\tilde{p}_t$, MC (non-adaptive), and MA+reg (non-adaptive) show minimal adaptivity to the underlying shifts and are expected to perform poorly across some subgroups over local intervals; whereas, our proposed algorithm has significantly better local multiaccuracy. While the locally adaptive MC algorithm improves adaptivity relative to non-adaptive

MC, its multiaccuracy rate is substantially worse than that of MA+reg (locally adaptive). Notably, it also has higher multiaccuracy error than MA+reg (non-adaptive) on some local intervals. We note that while MA (locally adaptive) performs slightly better in terms of multiaccuracy compared to MA+reg (locally adaptive), it suffers from significantly higher regret over all local intervals as can be seen from the right plot in Figure 3.

### 5.4 Comparison with adaptive multicalibration method

Finally, we compare our algorithm with an adaptive extension of the online multicalibration algorithm proposed in Lee et al. (2022) (MC (adaptive)) in Figures 4 and 5. This algorithm guarantees low multicalibration error on all subintervals in $[T]$ at the expense of higher runtime and memory as discussed in Section 2.3. While we use the fixed width values $\tau = 336$ for GEFCom2014-L and $\tau = 50$ for COMPAS in our algorithm, we perform a general evaluation here over different interval widths $|I|$. We find that while adaptivity improves the performance of the multicalibration algorithm, MA+reg (locally adaptive) still has significantly better local multiaccuracy across all interval widths on both datasets. In Appendix F.1, we show quantitative results for a wide range of window sizes $|I|$.

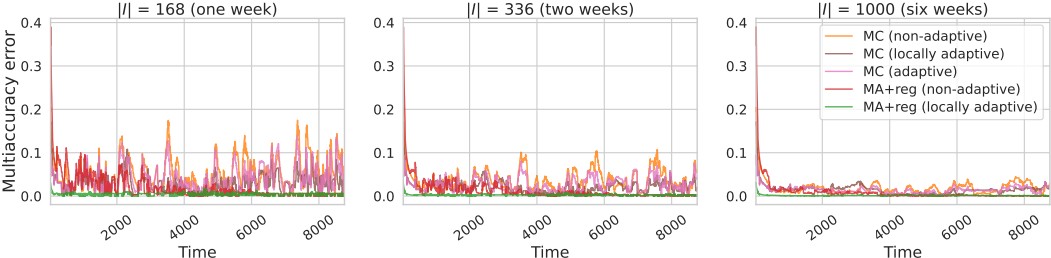

Figure 4: Local multiaccuracy error on GEFCom2014-L for different interval widths. We skip the first thirty time steps when plotting the multiaccuracy error for improved readability.

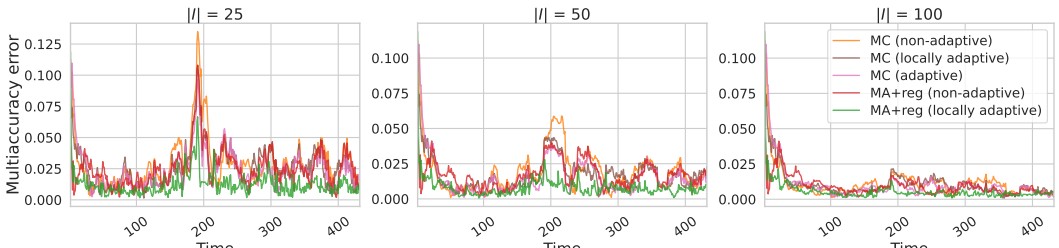

Figure 5: Local multiaccuracy error on COMPAS for different interval widths. We skip the first two time steps when plotting the multiaccuracy error for improved readability.

## 6 Discussion

We present adaptive multi-objective learning algorithms that guarantee small error for all objectives over local time intervals. In this growing literature, we hope our work serves as an initial step toward bridging the empirical gap. We note two limitations of our work: firstly, our guarantees hold over intervals of fixed width. Developing efficient algorithms that provide stronger guarantees in this setting remains an important problem. Second, our empirical evaluation focuses on a subset of objectives and validation on broader problems is interesting future work.

### Reproducibility Statement

We provide the code to run our algorithms and reproduce our experiments as part of the supplementary material. We state the assumptions in the main text and include proofs for our theoretical results in Appendix B. We describe all experimental details in Section 5.

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

## A EXTENSIONS

We now discuss the several extensions of our general algorithm.

**Quantile estimation.** Analogous to mean estimation, our algorithm can be used to update predicted quantiles to satisfy group-conditional coverage guarantees while preserving the quantile loss $\ell_\alpha$ (also referred to as pinball loss) with respect to baseline quantile predictions. We provide the full algorithm in Algorithm 5. Note that we have to allow quantile predictors $\theta_t$ to be random in this algorithm.

**Multi-group learning.** Our algorithm can be extended to multi-group learning with the set of objectives of the form $\mathbb{1}\{x_t \in g\}(\ell(p_t(x_t), y_t) - \ell(f(x_t), y_t))$ for groups $g \in \mathcal{G}$ and functions $f \in \mathcal{F}$.

**Omniprediction.** Omniprediction is a straightforward extension of our algorithm where the set of objectives are of the form $\ell(p_t(x_t), y_t) - \ell(f(x_t), y_t)$ for functions $f \in \mathcal{F}$ and losses $\ell \in \mathcal{L}$.

## B PROOFS

### B.1 PROOF OF LEMMA 1

We follow the calculations of Gradu et al. (2023). The primary difference between our work and these articles is that our losses may take on negative values.

Let $W^{(t+1)} := \sum_\ell w_\ell^{(t)} \exp\left(\eta \cdot \ell(p_t(x_t, x_t, y_t))\right)$. We initialize $w_\ell^{(t)} = 1$ for all $\ell \in \mathcal{L}$. By construction, the probabilities $q_\ell^{(t)} := \frac{w_\ell^{(t)}}{\sum_\ell w_\ell^{(t)}}$. Thus,

$$\frac{W^{(t+1)}}{W^{(t)}} = \sum_{\ell \in \mathcal{L}} q_\ell^{(t)} \exp\left(\eta \cdot \ell(p_t(x_t, x_t, y_t))\right).$$

Since $\eta$ is small and $\ell$ is bounded between $[-1, 1]$, $|\eta \cdot \ell(p_t(x_t, x_t, y_t)| \leq 1$. We use the inequalities $1 - a \leq \exp(-a)$ and for $|x| \leq 1$, $\exp(a) \leq 1 + a + a^2$ to get

$$\frac{W^{(t+1)}}{W^{(t)}} \leq \exp\left(\eta q^{(t)\top}\ell^{(t)} + \eta^2 q^{(t)\top}\ell_\mathcal{L}^{(t)2}\right).$$

This inductively implies, for interval $I = [r, s]$

$$\frac{W^{(s+1)}}{W^{(r)}} \leq \exp\left(\sum_{t=r}^{s} \eta q^{(t)\top}\ell^{(t)} + \eta^2 q^{(t)\top}\ell_\mathcal{L}^{(t)2}\right).$$

On the other hand, for any fixed $\ell \in \mathcal{L}$, $w_\ell^{(t+1)} \geq w_\ell^{(t)}(1 - \gamma)\exp\left(\eta\ell^{(t)}\right)$. Without loss of generality, we proceed with a fixed $\ell$, noting that the same calculations will follow for all $\ell \in \mathcal{L}$. This gives

$$\frac{W^{(s+1)}}{W^{(r)}} \geq \frac{w_\ell^{(s+1)}}{W^{(r)}} \geq (1 - \gamma)^{|I|}q_\ell^{(t)} \exp\left(\sum_{t=r}^{s} \eta\ell^{(t)}\right)$$

$$\geq (1 - \gamma)^{|I|}\frac{\gamma}{|\mathcal{L}|} \exp\left(\sum_{t=r}^{s} \eta\ell^{(t)}\right)$$

Combining the two inequalities and taking logarithm on both sides yields

$$|I|(1 - \gamma) + \log\left(\frac{\gamma}{|\mathcal{L}|}\right) + \sum_{t=r}^{s} \eta\ell^{(t)} \leq \sum_{t=r}^{s} \eta q^{(t)\top}\ell_\mathcal{L}^{(t)} + \eta^2 q^{(t)\top}\ell_\mathcal{L}^{(t)2}.$$

We rearrange to get the following inequality

$$\sum_{t=r}^{s} q^{(t)\top}\ell_\mathcal{L}^{(t)} \geq \sum_{t=r}^{s} \ell^{(t)} - \eta\left(\sum_{t=r}^{s} q^{(t)\top}\ell_\mathcal{L}^{(t)2}\right) + \frac{1}{\eta}|I|(1 - \gamma) + \log\left(\frac{\gamma}{|\mathcal{L}|}\right).$$

As $\gamma \leq 1/2$, we can use the inequality $\log(1 - \gamma) \geq -2\gamma$ to get the final inequality

$$\sum_{t=r}^{s} q^{(t)\top} \ell_{\mathcal{L}}^{(t)} \geq \sum_{t=r}^{s} \ell^{(t)} - \eta \left( \sum_{t=r}^{s} q^{(t)\top} \ell_{\mathcal{L}}^{(t)^2} \right) - \frac{1}{\eta} \left( \log \left( \frac{|\mathcal{L}|}{\gamma} \right) + |I| 2\gamma \right).$$

As the same calculation holds for any objective $\ell \in \mathcal{L}$, we get the final result

$$\sum_{t=r}^{s} q^{(t)\top} \ell_{\mathcal{L}}^{(t)} \geq \max_{\ell \in \mathcal{L}} \sum_{t=r}^{s} \ell^{(t)} - \eta \left( \sum_{t=r}^{s} q^{(t)\top} \ell_{\mathcal{L}}^{(t)^2} \right) - \frac{1}{\eta} \left( \log \left( \frac{|\mathcal{L}|}{\gamma} \right) + |I| 2\gamma \right).$$

### B.2 PROOF OF LEMMA 2

This result was shown in Lee et al. (2022) and we follow the same calculations.

Let $u^{(t)}(P, y) := \sum_{\ell} q_{\ell}^{(t)} \mathbb{E}_{p \sim P}[\ell(p, x_t, y_t)]$. Let $\Delta(\mathcal{Y})$ denote the space of distributions over $\mathcal{Y}$. Applying Sion's Minimax Theorem, we get

$$\min_{P \in \Delta(\mathcal{Y})} \max_{y \in \mathcal{Y}} u^{(t)}(P, y) = \min_{P \in \Delta(\mathcal{Y})} \max_{P_Y \in \Delta(\mathcal{Y})} \mathbb{E}_{y \sim P_Y}[u^{(t)}(P, y)]$$

$$= \max_{P_Y \in \Delta(\mathcal{Y})} \min_{P \in \Delta(\mathcal{Y})} \mathbb{E}_{y \sim P}[u^{(t)}(P, y)].$$

In particular, we find that the minimax-optimal strategy $P_t(x_t)$ of the learner can achieve a value $u^{(t)}(P_t(x_t), y)$ as low as if the distribution of $y_t$ was chosen first and the learner could best-respond. In this latter case, we have by Assumption 1 that there exists $p^*$ such that $u^{(t)}(p^*, y) \leq 0$.

Thus, the minimax optimal strategy guarantees that $\min_{P \in \Delta(\mathcal{Y})} \max_{y \in \mathcal{Y}} u^{(t)}(P, y) \leq 0$ for all $t \in [T]$. This yields the desired inequality

$$\sum_{t=r}^{s} q^{(t)\top} \ell_{\mathcal{L}}^{(t)} \leq 0.$$

### B.3 PROOF OF THEOREM 2

Applying Lemma 2 to the inequality (6) in Lemma 1 gives

$$\max_{\ell \in \mathcal{L}} \sum_{t=r}^{s} \ell^{(t)} - \eta \left( \sum_{t=r}^{s} q^{(t)\top} \ell_{\mathcal{L}}^{(t)^2} \right) - \frac{1}{\eta} \left( \log \left( \frac{|\mathcal{L}|}{\gamma} \right) + |I| 2\gamma \right) \leq 0.$$

Rearranging and dividing both sides by $|I|$ yields the desired inequality,

$$\max_{\ell \in \mathcal{L}} \frac{1}{|I|} \sum_{t=r}^{s} \ell^{(t)} \leq \frac{\eta}{|I|} \left( \sum_{t=r}^{s} q^{(t)\top} \ell_{\mathcal{L}}^{(t)^2} \right) + \frac{1}{\eta |I|} \left( \log \left( \frac{|\mathcal{L}|}{\gamma} \right) + |I| 2\gamma \right).$$

### B.4 PROOF OF COROLLARY 1

The proof follows by instantiating the set of objectives $\mathcal{L}$ for multiaccurate mean estimation in Theorem 2. We take $\mathcal{L} := \{\ell_{\mathrm{MA}_{f,\sigma}} : f \in \mathcal{F}_{\mathrm{MA}}, \sigma \in \{\pm\}\} \cup \{\ell_{\mathrm{pred}}\}$, where $\mathcal{F}_{\mathrm{MA}} \subseteq \{f : \mathcal{X} \to [0,1]\}$ is the function class that we desire multiaccuracy with respect to and $\ell_{\mathrm{pred}}$ is the prediction error objective. Plugging the objectives in equation 8, this gives us the desired bound

$$\max \left\{ \max_{f,\sigma} \frac{1}{|I|} \sum_{t=r}^{s} \ell_{\mathrm{MA}_{f,\sigma}}^{(t)}, \frac{1}{|I|} \sum_{t=r}^{s} \ell_{\mathrm{pred}}^{(t)} \right\} = O\left( \sqrt{\frac{\log(|\mathcal{L}| \cdot |I|)}{|I|}} \right).$$

## C DEFERRED ALGORITHMS

In Section 4.1, we discussed the significance of the prediction error objective in preserving the accuracy relative to a base predictor sequence $\tilde{p}_t$. When $\tilde{p}_t$ is not available in advance, we can

combine our procedure with a standard online learning algorithm (e.g., online gradient or mirror descent) that provides an appropriate baseline. Algorithm 3 gives a complete description of this approach. In what follows, the weights $q_{\mathrm{MA},f,\sigma}^{(t)}$ and $q_{\mathrm{reg}}^{(t)}$ are used to denote the entries of $q^{(t)}$ associated with the multiaccuracy and regret objectives, respectively.

---

**Algorithm 3** Locally adaptive multiaccurate mean estimation (learning $\tilde{p}_t$ online)

---

**Input:** Function class $\mathcal{F}_{\mathrm{MA}} \subseteq \{f : \mathcal{X} \to [0,1]\}$; $\mathcal{F}_{\mathrm{pred}} = \{f_\beta : \beta \in \mathbb{R}\}$; hyperparameters $\eta, \gamma$.
**Input:** Sequence of samples $\{(x_1, y_1), \ldots, (x_T, y_T)\}$

1: $q_{\mathrm{MA}_{f,\sigma}}^{(1)} = \frac{1}{2|\mathcal{F}_{\mathrm{MA}}|+1}, \quad \forall f \in \mathcal{F}_{\mathrm{MA}}, \sigma \in \{\pm 1\}$.

2: $q_{\mathrm{reg}}^{(1)} = \frac{1}{2|\mathcal{F}_{\mathrm{MA}}|+1}$

3: $\beta_1 = 0$

4: **for** each $t \in [T]$ **do**

5: $\quad \beta_{t+1} = \beta_t - \gamma \nabla_\beta \ell\left(f_{\beta_t}(x_t), y_t\right)$

6: $\quad \tilde{p}_t(x_t) := f_{\beta_t}(x_t)$

7: $\quad p_t(x_t) := \underset{p}{\operatorname{argmin}} \max_{y \in \{0,1\}} \sum_{f,\sigma} q_{\mathrm{MA}_{f,\sigma}}^{(t)} \sigma f(x_t)(y - p(x_t)) + q_{\mathrm{reg}}^{(t)}(\ell_{\mathrm{acc}}(p(x_t), y) - \ell_{\mathrm{acc}}(\tilde{p}_t(x_t), y))$

8: $\quad \tilde{q}_{\mathrm{MA}_{f,\sigma}}^{(t+1)} \propto q_{\mathrm{MA}_{f,\sigma}}^{(t)} \exp\left(\eta \cdot \sigma f(x_t)(y_t - p_t(x_t))\right)$ for all $f \in \mathcal{F}, \sigma \in \{+,-\}$

9: $\quad \tilde{q}_{\mathrm{reg}}^{(t+1)} \propto q_{\mathrm{reg}}^{(t)} \exp\left(\eta \cdot (\ell(p_t(x_t), y_t) - \ell(\tilde{p}_t(x_t), y_t))\right)$

10: $\quad q_{\mathrm{MA}_{f,\sigma}}^{(t+1)} = (1 - \gamma) \tilde{q}_{\mathrm{MA}_{f,\sigma}}^{(t+1)} + \frac{\gamma}{2|\mathcal{F}_{\mathrm{MA}}|+1}$

11: $\quad q_{\mathrm{reg}}^{(t+1)} = (1 - \gamma) \tilde{q}_{\mathrm{reg}}^{(t+1)} + \frac{\gamma}{2|\mathcal{F}_{\mathrm{MA}}|+1}$

**Output:** Sequence of (randomized) predictors $p_1, \ldots, p_T$

---

Algorithms 4 and 5 give specific instantiations of Algorithm 2 for locally-adaptive multiaccurate mean and quantile estimation respectively.

---

**Algorithm 4** Locally adaptive multiaccurate mean estimation

---

**Input:** Function class $\mathcal{F}_{\mathrm{MA}} \subseteq \{f : \mathcal{X} \to [0,1]\}$; base predictor sequence $\tilde{p}_t, t \in [T]$; hyperparameters $\eta, \gamma$.
**Input:** Sequence of samples $\{(x_1, y_1), \ldots, (x_T, y_T)\}$

1: $q_{\mathrm{MA}_{f,\sigma}}^{(1)} = \frac{1}{2|\mathcal{F}_{\mathrm{MA}}|+1}, \quad \forall f \in \mathcal{F}_{\mathrm{MA}}, \sigma \in \{\pm 1\}$.

2: $q_{\mathrm{reg}}^{(1)} = \frac{1}{2|\mathcal{F}_{\mathrm{MA}}|+1}$

3: **for** each $t \in [T]$ **do**

4: $\quad p_t(x_t) := \underset{p}{\operatorname{argmin}} \max_{y \in \mathcal{Y}} \sum_{f,\sigma} q_{\mathrm{MA}_{f,\sigma}}^{(t)} \sigma f(x_t)(y - p(x_t)) + q_{\mathrm{reg}}^{(t)}(\ell_{\mathrm{acc}}(p(x_t), y) - \ell_{\mathrm{acc}}(\tilde{p}_t(x_t), y))$

5: $\quad \tilde{q}_{\mathrm{MA}_{f,\sigma}}^{(t+1)} \propto q_{\mathrm{MA}_{f,\sigma}}^{(t)} \exp\left(\eta \cdot \sigma f(x_t)(y_t - p_t(x_t))\right)$ for all $f \in \mathcal{F}_{\mathrm{MA}}, \sigma \in \{\pm 1\}$

6: $\quad \tilde{q}_{\mathrm{reg}}^{(t+1)} \propto q_{\mathrm{reg}}^{(t)} \exp\left(\eta \cdot (\ell_{\mathrm{acc}}(p_t(x_t), y_t) - \ell_{\mathrm{acc}}(\tilde{p}_t(x_t), y_t))\right)$

7: $\quad q_{\mathrm{MA}_{f,\sigma}}^{(t+1)} = (1 - \gamma) \tilde{q}_{\mathrm{MA}_{f,\sigma}}^{(t+1)} + \frac{\gamma}{2|\mathcal{F}_{\mathrm{MA}}|+1}$

8: $\quad q_{\mathrm{reg}}^{(t+1)} = (1 - \gamma) \tilde{q}_{\mathrm{reg}}^{(t+1)} + \frac{\gamma}{2|\mathcal{F}_{\mathrm{MA}}|+1}$

**Output:** Sequence of (randomized) predictors $p_1, \ldots, p_T$

---

# D ADDITIONAL EXPERIMENTAL PARTICULARS

## D.1 GEFCom2014 ELECTRIC LOAD FORECASTING

For our electric load forecasting experiment, we need to compute the hourly probability that electricity demand exceeds a threshold (150 MW in our example) given quantile forecasts. We use linear interpolation to estimate the full cumulative distribution function of the load from the quantile forecasts of Ziel & Liu (2016). Their method outperforms top entries of the competition. Fix a set of quantile levels $0 < \alpha_1 < \cdots < \alpha_k$ and let the corresponding set of quantile forecasts at hour $t$

---

**Algorithm 5** Locally adaptive multiaccurate quantile estimation

---

**Input:** Function class $\mathcal{F}_{\text{MA}} \subseteq \{f : \mathcal{X} \to [0,1]\}$; quantile level $\alpha$; quantile predictions $\tilde{\theta}_t, t \in [T]$; hyperparameters $\eta, \gamma$.
**Input:** Sequence of samples $\{(x_1, y_1), \ldots, (x_T, y_T)\}$

1: $q^{(1)}_{\text{MA}_{f,\sigma}} = \frac{1}{2|\mathcal{F}_{\text{MA}}|+1}, \quad \forall f \in \mathcal{F}_{\text{MA}}, \sigma \in \{\pm 1\}$.

2: $q^{(1)}_{\text{reg}} = \frac{1}{2|\mathcal{F}_{\text{MA}}|+1}$

3: **for** each $t \in [T]$ **do**

4: $\quad \theta_t(x_t) := \underset{\theta \in \Delta(\Theta)}{\operatorname{argmin}} \max_{y \in \mathcal{Y}} \sum_{f,\sigma} q^{(t)}_{\text{MA}_{f,\sigma}} \sigma f(x_t)(\mathbb{1}\{y \leq \theta\} - \alpha) + q^{(t)}_{\text{reg}}\left(\ell_\alpha(\theta, y) - \ell_\alpha(\tilde{\theta}_t, y_t)\right)$

5: $\quad \tilde{q}^{(t+1)}_{\text{MA}_{f,\sigma}} \propto q^{(t)}_{\text{MA}_{f,\sigma}} \exp\left(\eta \cdot \sigma \cdot f(x_t)(\mathbb{1}\{y_t \leq \theta_t(x_t)\} - \alpha)\right)$ for all $f \in \mathcal{F}_{\text{MA}}, \sigma \in \{\pm 1\}$

6: $\quad \tilde{q}^{(t+1)}_{\text{reg}} \propto q^{(t)}_{\text{reg}} \exp\left(\eta \cdot (\ell_\alpha(\theta_t(x_t), y_t) - \ell_\alpha(\tilde{\theta}_t, y_t))\right)$

7: $\quad q^{(t+1)}_{\text{MA}_{f,\sigma}} = (1-\gamma)\tilde{q}^{(t+1)}_{\text{MA}_{f,\sigma}} + \frac{\gamma}{2|\mathcal{F}_{\text{MA}}|+1}$

8: $\quad q^{(t+1)}_{\text{reg}} = (1-\gamma)\tilde{q}^{(t+1)}_{\text{reg}} + \frac{\gamma}{2|\mathcal{F}_{\text{MA}}|+1}$

**Output:** Sequence of (randomized) quantile predictors $\theta_1, \ldots, \theta_T$

---

be $\hat{\theta}^{\alpha_1}_t < \cdots < \hat{\theta}^{\alpha_k}_t$. Let $Y_t \in \mathbb{R}$ denote the hourly load. We estimate the cumulative distribution function of $Y$ by linearly interpolating between the points $\{(\hat{\theta}^{\alpha_i}_t, \alpha_i)\}^k_{i=1}$. Formally, for any $x \in \mathbb{R}$

$$\widehat{\mathbb{P}}(Y \leq x) = \begin{cases} 0, & x < \alpha_1, \\ 1, & x \geq \alpha_k, \\ \alpha_{i-1} + \frac{\alpha_i - \alpha_{i-1}}{\hat{\theta}^{\alpha_i}_t - \hat{\theta}^{\alpha_{i-1}}_t}\left(x - \hat{\theta}^{\alpha_{i-1}}_t\right), & \hat{\theta}^{\alpha_{i-1}}_t \leq x < \hat{\theta}^{\alpha_i}_t. \end{cases}$$

Figure 6 shows the constructed baseline predictions $\tilde{p}_t$ for the task using the above procedure along with the raw load variable.

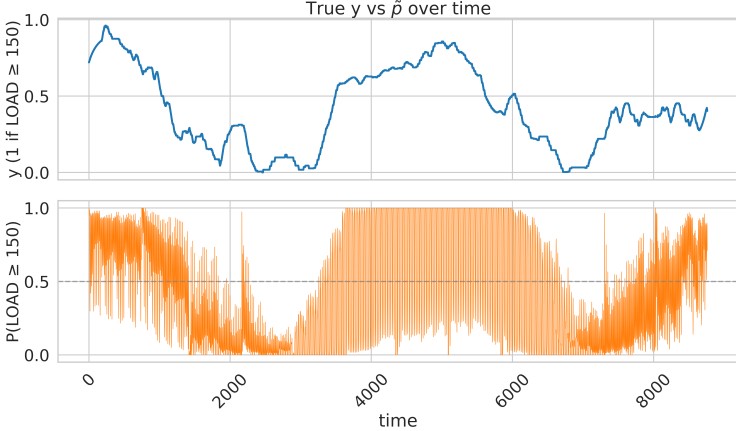

Figure 6: **GEFCom2014-L: True load $(y)$ and constructed predictions** $\tilde{p}_t$. We plot the moving average of the binary $y$ over a window size $|I| = 336$ hours (2 weeks) (top) and the baselines predictions $\tilde{p}_t$ constructed from the quantile forecasts using linear interpolation (bottom) over time.

### D.2 MULTICALIBRATION IMPLEMENTATION

We implement the multicalibration + calibeating algorithm in Lee et al. (2022) and calibeat the baseline forecaster sequence $\tilde{p}_t$. We set the optimal choice of $\eta = \sqrt{\frac{\log(2|\mathcal{L}|m)}{4T}}$ for the algorithm. We take the number of bins $m = 10$ and 10 level sets of the forecaster throughout. Figure 7 shows the

total multiaccuracy error and regret over all intervals for varying values of $m$. As $m$ decreases, the multicalibration algorithm approaches the multiaccuracy algorithm and the total MA error decreases. Even when $m = 2$, MA+reg has lower total MA error and regret than MC on GEFCom2014-L (Figure 7a). While the total MA error of MC drops below MA+reg on COMPAS with smaller $m$ (Figure 7b), this is accompanied by an increase in total regret.

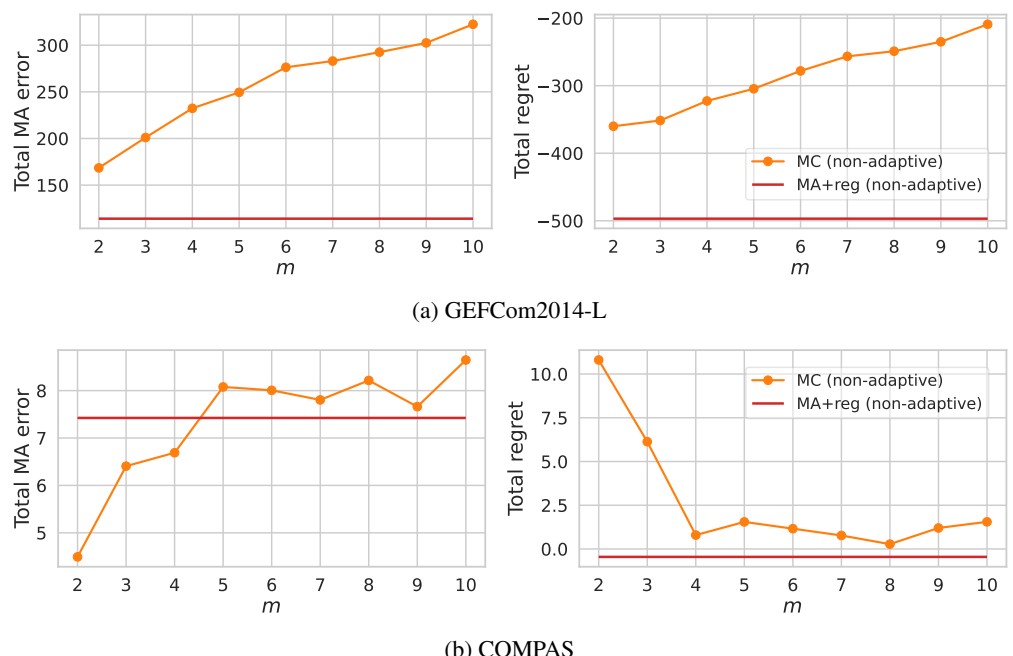

(a) GEFCom2014-L

(b) COMPAS

Figure 7: **Total multiaccuracy error and regret with varying $m$,** (a) GEFCom2014-L and (b) COMPAS. This is the same setting as Figure 2 and Figure 3 where we now vary the number of bins $m$.

## E  COMPARISON WITH ADAPTIVE MA+REG

In Section 5.4, we compared our proposed locally adaptive MA+reg algorithm with the adaptive online multicalibration algorithm proposed in Lee et al. (2022). Now, we use the adaptive method proposed in Lee et al. (2022) with the MA+reg objectives. See Figures 8 and 9 for the results, where the algorithm is labeled as MA+reg (adaptive). Results show that while MA+reg (adaptive) improves the multiaccuracy error over the non-adaptive baseline, it is consistently outperformed by MA+reg (locally adaptive) in all settings. This comparison shows that even when the adaptive baseline has the same objectives, the locally adaptive algorithm exceeds its performance.

## F  ABLATIONS ON HYPERPARAMETERS

### F.1  VARYING INTERVAL WIDTH $|I|$

We extend the analysis in Section 5.4 and plot the total multiaccuracy error over all windows for a wide range of varying interval widths $|I|$. Results in Figure 10 show that locally adaptive MA+reg consistently outperforms all other adaptive algorithms despite being tuned with a fixed width. While MC (locally adaptive) improves upon the non-adaptive MC algorithm, the multiaccuracy error remains significantly higher than MA+reg (locally adaptive). It is interesting to note that MC (adaptive) does not achieve lower total multiaccuracy error than MC (locally adaptive) despite its stronger theoretical guarantee over all subintervals.

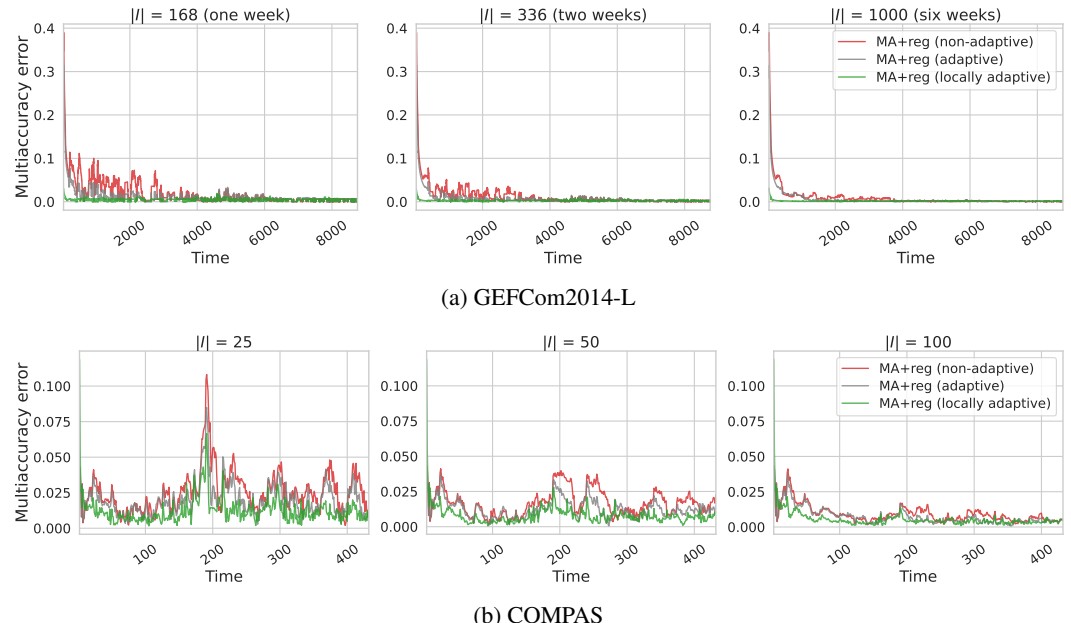

Figure 8: **Local multiaccuracy error for different interval widths $|I|$,** (a) GEFCom2014-L and (b) COMPAS. This is the same setting as Figure 4 and Figure 5 where we now show comparison with the adaptive MA+reg algorithm.

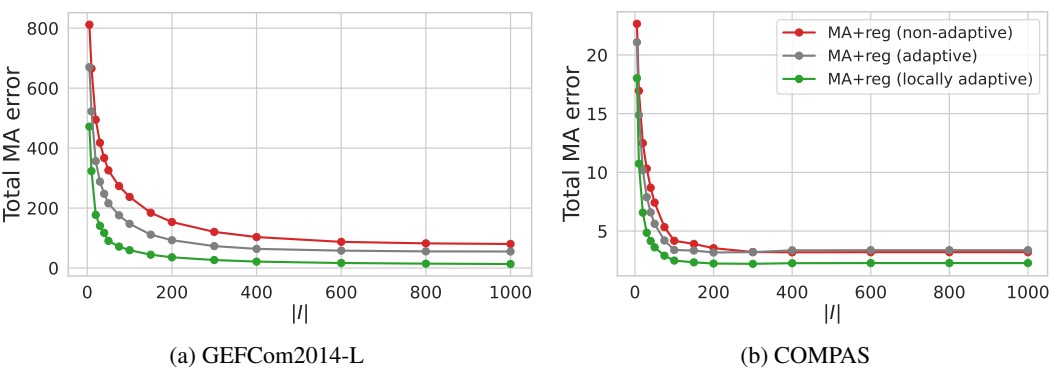

Figure 9: **Total multiaccuracy error with varying interval width $|I|$,** (a) GEFCom2014-L and (b) COMPAS. This is the same setting as Figure 4 and Figure 5. We vary the window width $|I|$ used for the moving average of errors and plot the total multiaccuracy error under the curve.

### F.2 VARYING $\eta$

In this section, we consider three different choices of $\eta$ in the locally adaptive MA+reg algorithm.

1. $\eta = \sqrt{\frac{\log |\mathcal{L}|}{T}}$: this is the optimal $\eta$ used in the non-adaptive variant.

2. $\eta = \sqrt{\frac{\log((2|\mathcal{F}_{\mathrm{MA}}|+1)\cdot 2\tau)+1}{\tau}}$: we substitute the online updates $\sum_{s=t-\tau+1}^{t} q_{\mathrm{MA}}^{(s)\top} \ell_{\mathrm{MA}}^{(s)^2} + q_{\mathrm{reg}}^{(s)} \ell_{\mathrm{reg}}^{(s)^2}$ in the adaptive choice of $\eta_t$ (9) with the interval width $\tau$.

3. $\eta = \eta_t := \sqrt{\frac{\log((2|\mathcal{F}_{\mathrm{MA}}|+1)\cdot 2\tau)+1}{\sum_{s=t-\tau+1}^{t} q_{\mathrm{MA}}^{(s)\top} \ell_{\mathrm{MA}}^{(s)^2} + q_{\mathrm{reg}}^{(s)} \ell_{\mathrm{reg}}^{(s)^2}}}$: this is the adaptive choice of $\eta$ proposed in (9), which is the default for our algorithm.

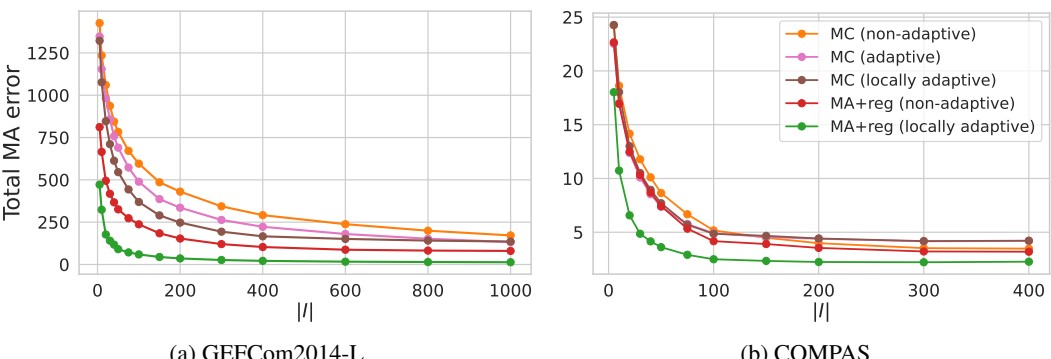

(a) GEFCom2014-L       (b) COMPAS

Figure 10: **Total multiaccuracy error with varying interval width $|I|$,** (a) GEFCom2014-L and (b) COMPAS. This is the same setting as Figure 4 and Figure 5. We vary the window width $|I|$ used for the moving average of errors and plot the total multiaccuracy error under the curve.

See Figures 11 and 12 for the results that show the local multiaccuracy error and total multiaccuracy error respectively with the above choices of $\eta$ and varying interval widths. Adaptive $\eta_t$ consistently dominates, followed by the the choice of $\eta$ that uses interval width $\tau$. These results shows that the exploration term alone is not sufficient to achieve local adaptivity.

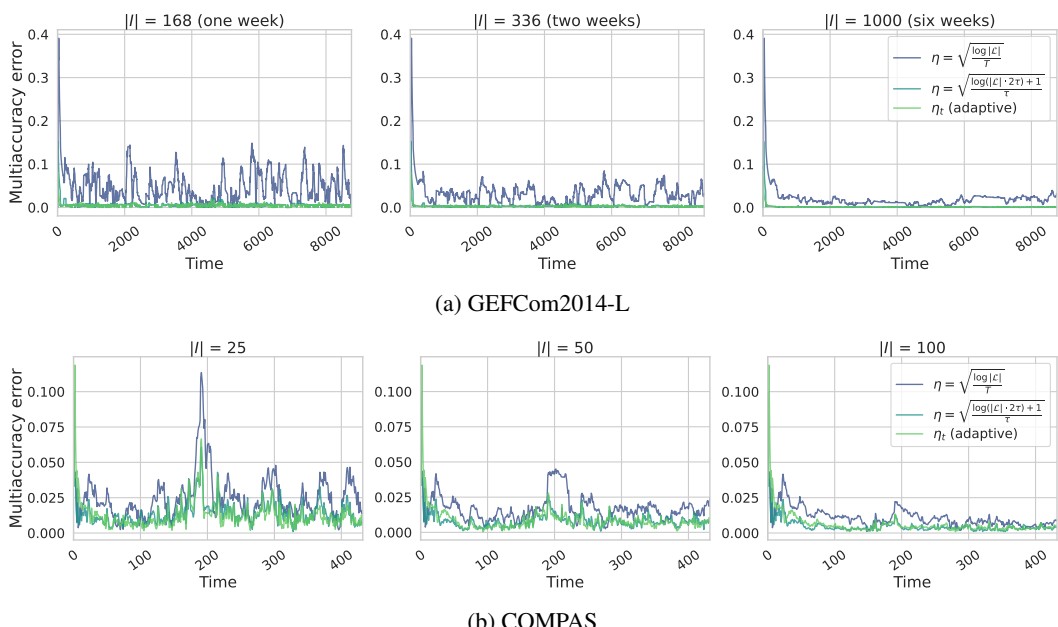

(a) GEFCom2014-L

(b) COMPAS

Figure 11: **Local multiaccuracy error with varying $\eta$ for different interval widths $|I|$,** (a) GEFCom2014-L and (b) COMPAS. This is the same setting as Figure 4 and Figure 5 where we now show results with different choices of $\eta$ in the locally adaptive MA+reg algorithm.

### F.3 VARYING $\tau$ AND $\gamma$

We now vary the fixed interval width $\tau$ used for tuning the locally adaptive MA+reg algorithm. This also results in different values of optimal $\gamma = 1/2\tau$. We evaluate the total multiaccuracy error for different choices of $\tau$ over windows of varying width $|I|$ in Figure 13. Results show that the total error does not significantly change with different $\tau$ values and that the locally adaptive algorithm is robust to the choice of $\tau$.

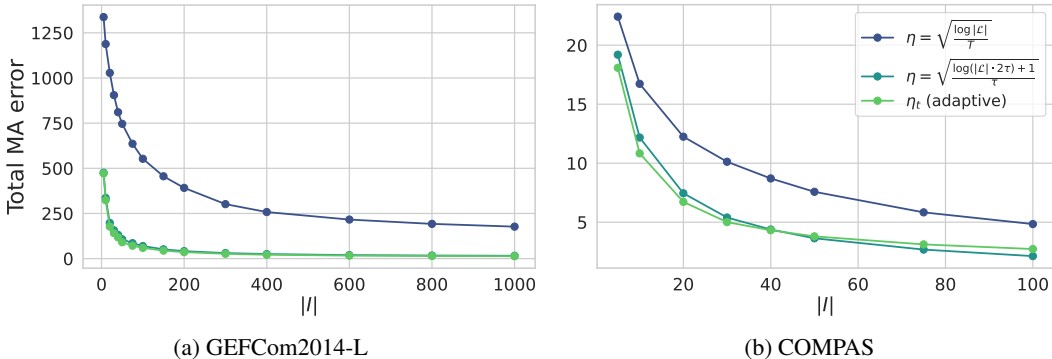

(a) GEFCom2014-L        (b) COMPAS

Figure 12: **Total multiaccuracy error with varying $\eta$ and interval width $|I|$,** (a) GEFCom2014-L and (b) COMPAS. This is the same setting as Figure 4 and Figure 5. We vary the window width $|I|$ used for the moving average of errors and plot the total multiaccuracy error under the curve with different choices of $\eta$ in the locally adaptive MA+reg algorithm.

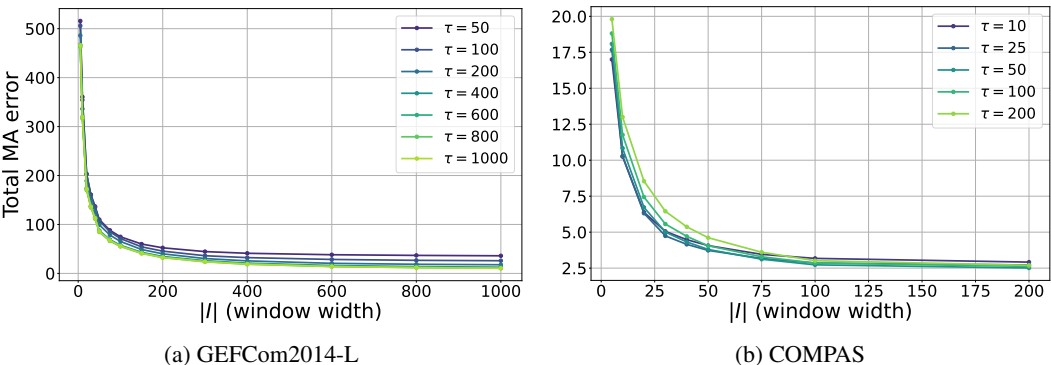

(a) GEFCom2014-L        (b) COMPAS

Figure 13: **Total multiaccuracy error for different $\tau$ with varying interval width $|I|$,** (a) GEFCom2014-L and (b) COMPAS. This is the same setting as Figure 4 and Figure 5. We vary the fixed width $\tau$ used for tuning MA+reg (locally adaptive) and the window width $|I|$ used for the moving average of errors and plot the total multiaccuracy error under the curve.

## G  SIMULATED EXAMPLES

We consider a set of simulated examples where we can control the distribution shifts over time. We focus on a simple setting with a time-varying linear model

$$Y_t = X_t^\top \beta_t + \varepsilon_t, \qquad \varepsilon_t \sim \mathcal{N}(0, \sigma^2),$$

where the covariates $X_t$ i.i.d. Gaussian,

$$X_t \sim \mathcal{N}(0, I_d),$$

and we specify the distribution shift entirely through the coefficients $\beta_t \in \mathbb{R}^d$. The initial $\beta_0 \sim \mathcal{N}\left(0, \frac{1}{d} I_d\right)$ and we set

$$\beta_t = \beta_0 + \mu_t\, v,$$

where $v \in \mathbb{R}^d$ is a unit direction vector and $(\mu_t)_{t=1}^T$ controls the magnitude of the shift along direction $v$. We consider *jump* discontinuities in $\mu_t$ of varying sizes. Specifically, we divide the time horizon into three equally sized intervals and define $\mu_t$ to have small-amplitude jumps in the first and third intervals and large-amplitude jumps in the second interval. We construct three jump–shift datasets (*small, medium*, and *large*) by increasing the small-amplitude range and the large-amplitude range of $\mu_t$ as

- s*mall* shift: $\mu_t$ oscillates between $[-0.05, 0.05]$ in the small-amplitude regime and between $[-0.5, 0.5]$ in the large-amplitude regime.

- *medium* shift: $\mu_t$ oscillates between $[-0.075, 0.075]$ in the small-amplitude regime and between $[-1.0, 1.0]$ in the large-amplitude regime.
- *large* shift: $\mu_t$ oscillates between $[-0.1, 0.1]$ in the small-amplitude regime and between $[-1.5, 1.5]$ in the large-amplitude regime.

We take $d = 5$ in our experiments. Figure 14 shows the final trajectories of $\mu_t$ and $\beta_{t,0}$, the first coordinate of $\beta_t$, in all three jump shift settings.

We set function class $\mathcal{F}$ to be all the covariates such that $f_j(X) = X_j$, $j \in [d]$. Figure 15 shows the results comparing all algorithms across the three settings. While all algorithms reasonably adapt to the distribution shift in the small jump shift setting, MA+reg (locally adaptive) consistently outperforms all methods as the magnitude of shift increases. The difference is especially substantial in the large jump shift setting. Results from these examples show that the proposed locally adaptive algorithm is able to adapt to discontinuous and abrupt distribution shifts with better rates than existing methods.

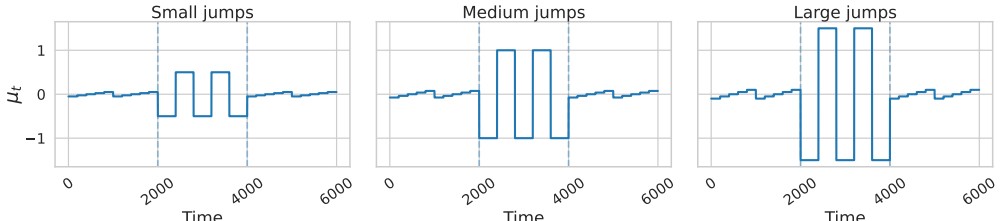

(a) Trajectory for $\mu_t$ in the *small* (left), *medium* (mid), and *large* (right) jump shift settings.

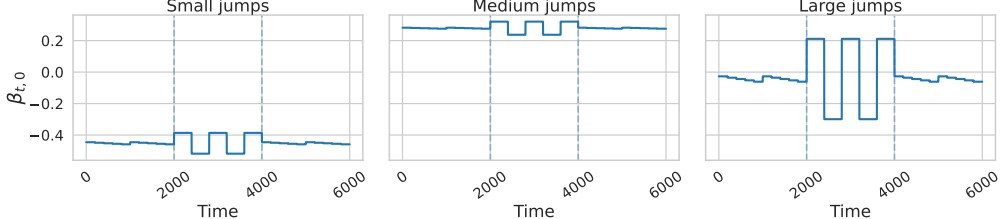

(b) Trajectory for $\beta_{t,0}$ in the *small* (left), *medium* (mid), and *large* (right) jump shift settings.

Figure 14: **Trajectories for (a) $\mu_t$ and (b) $\beta_{t,0}$ in the different jump shift settings.** $\beta_{t,0}$ denotes the first coordinate of $\beta_t$. $\beta_{t,j}$ is an affine transformation of $\mu_t$ by construction. Dashed vertical lines denote the boundaries between the regime switches where the size of the distribution shift changes.

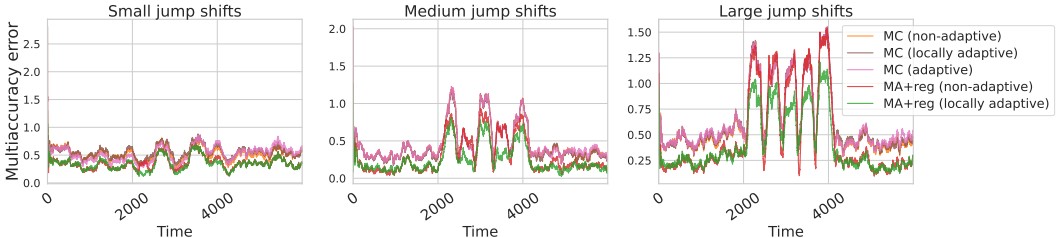

Figure 15: **Local multiaccuracy error in different jump shift settings,** *small* (left), *medium* (mid), and *large* (right) jump shifts.

# H    COMPAS DATASET

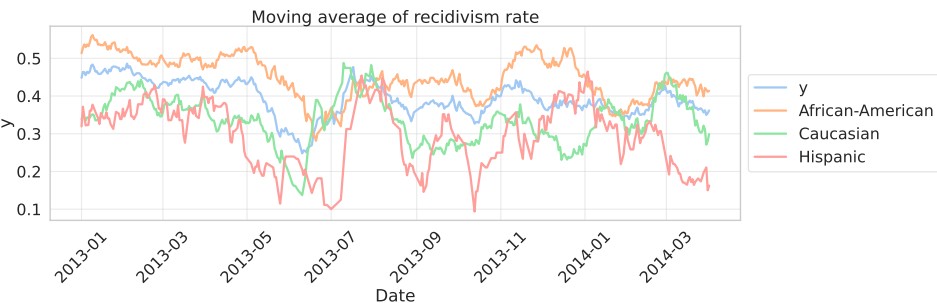

Figure 16: **COMPAS dataset.** Rolling average of true recidivism over time.

