# OpenReview forum: "Locally Adaptive Multi-Objective Learning"
_ICLR.cc/2026/Conference — Submitted to ICLR 2026_

### Official Review · Reviewer_L6nz · 2025-10-30

**Soundness:** 2
**Presentation:** 3
**Contribution:** 2
**Rating:** 4
**Confidence:** 3

**Summary:**

This paper investigates online multi-objective learning, aiming to learn a predictor that satisfies multiple objectivtes of interest simultaneously. The authors work in an online setting where the
data distribution can change arbitrarily over time. In the online setting, existing approaches to this problem that minimize the set of objectives over the entire time horizon can fail to adapt to distribution shifts. This paper proposes algorithms that guarantee small error for all objectives
over any local time interval of a given width.

**Strengths:**

The topic studied in this paper is interesting, and the authors have provided a thorough discussion of the related work.

**Weaknesses:**

I have outlined the main weaknesses of this paper in the Question section.

**Questions:**

According to the proposed Algorithm 1, I believe that this work does not actually address changing environments as claimed by the authors. In other words, referring to Algorithm 1 as an adaptive algorithm is not appropriate. Specifically, the authors describe the notion of strong adaptivity in the paper. It is worth noting that a strongly adaptive algorithm (Daniely et al., 2015) does not require prior knowledge of interval information and can automatically adapt to all unknown intervals.

However, the proposed Algorithm 1 requires interval information to set $\eta$. In my view, this is unreasonable, as it implies that once the environment changes, the authors would need to manually adjust the step size of the algorithm.

Please clarify this point.

---

> ### Author Response · Authors · 2025-11-24
> **Author Response**
>
> We thank the reviewer for their question. We respectfully disagree with the assertion that the proposed algorithm is not an adaptive algorithm.  We agree the algorithm is not parameter-free, however our extensive empirical evaluations (Figures 4, 5, and 10) show that the algorithm is not sensitive to the choice of this interval width and obtains better adaptive performance over local time intervals than a variety of alternatives. Results demonstrating the robustness of the proposed method to the values of the hyperparameters are given in the new experiments in Appendix F.
>
> We hope our response addresses the concern sufficiently. Please let us know if there is anything more we can do to further address your concerns or provide additional clarity.

---

> > ### Comment · Reviewer_L6nz · 2025-11-24
> >
> > Thank you for your revised version and response. However, my concern still remains. In fact, I do not believe that the empirical studies can validate the theoretical significance of the results in this work, as the paper focuses on theoretical analysis.
> >
> > Therefore, I have substantial concerns regarding the theoretical guarantees of the proposed adaptive algorithm. Specifically, if your algorithm requires prior knowledge of the interval, it seems meaningless in theoretical terms. To clarify, given the interval $I=[r,s]$, one could directly apply a static version of an online multi-objective algorithm within that interval $I$ and still achieve the guarantee in (8) of Theorem 2 presented in your work. Is this correct?
> >
> > Please clarify this point.

---

> > > ### Author Response · Authors · 2025-11-25
> > >
> > > Thank you for your engagement. We believe the above claim is incorrect. A static version of the online multi-objective algorithm run within an interval will not yield any guarantees for overlapping intervals while our theory holds for overlapping intervals. We would also like to note that we require knowledge of the interval **length**, not the specific interval.
> > >
> > > We hope the clarification addresses your concern, and we are happy to discuss further.

---

> > > > ### Comment · Reviewer_L6nz · 2025-11-25
> > > >
> > > > My example refers specifically to the final bound in (8), not to other guarantees.
> > > >
> > > > Furthermore, please explain how the learning rate $\eta$ is set when only the interval length is known. It appears that it requires $\sum_{t=r}^s q^\top (\ell)^2$, which seems to necessitate knowledge of the specific interval. Could you clarify this point?
> > > >
> > > > I also notice that $\sum_{t=r}^s q^\top (\ell)^2$ is not practical, because it requires the knowledge of $q$ in advance. Therefore, you set a different learning rate in (9). But, this would result in the loss of theoretical guarantees. Right?

---

> > > > > ### Author Response · Authors · 2025-11-26
> > > > >
> > > > > Thank you for the clarifying questions.
> > > > >
> > > > > > My example refers specifically to the final bound in (8), not to other guarantees.
> > > > >
> > > > > The bound we provide in (8) is indeed over all overlapping intervals of length $|I|$ as we stated above.
> > > > >
> > > > > > please explain how the learning rate $\eta$ is set; I also notice that $\sum_{t=r}^{s} q^{(s)\top} ({\ell^{(s)}_\mathcal{L}})^2$ is not practical, because it requires the knowledge of $q$ in advance.
> > > > >
> > > > > We set an adaptive value of $\eta_t$ (eq. 9) that tracks changes in the moving average of the expected squared objective over the most recent $|I| = \tau$ time steps. $q_{\ell}^{(t)}$ over the past $\tau$ steps is known and hence $\sum_{s=t-\tau +1}^{t} q^{(s)\top} ({\ell^{(s)}_\mathcal{L}})^2$ can be computed.
> > > > >
> > > > > This is an estimation as the expected squared objective is unknown in practice, which we also note in our paper (lines 349-360). $\eta = \sqrt{\frac{\log(|\mathcal{L}|\cdot 2\tau) + 1}{\tau}}$ with the fixed interval width $\tau$ yields the desired theoretical guarantee in (8); empirically, it performs slightly worse than the adaptive choice (see Appendix F.2).

---

### Official Review · Reviewer_iN2x · 2025-10-31

**Soundness:** 3
**Presentation:** 3
**Contribution:** 2
**Rating:** 4
**Confidence:** 3

**Summary:**

The paper proposes a locally adaptive algorithm for online multi-objective learning, aiming to ensure small error for all objectives over any local time interval, rather than minimizing a time-averaged global objective. The algorithm builds on exponential-weights updates and Fixed Share ideas to handle temporal adaptivity. The work focuses primarily on the multiaccuracy problem, where the goal is to learn unbiased predictors under covariate shift, while preserving performance relative to a base predictor through an added regret objective. The authors also discuss extensions to multicalibration, multi-group learning, and omniprediction. Empirical results on electricity demand forecasting (GEFCom2014-L) and COMPAS recidivism prediction suggest improved “local adaptivity” compared to non-adaptive baselines. The authors claim their work bridges a gap between theoretical multi-objective learning and empirical application.

**Strengths:**

1. The motivation is strong as local adaptivity under distribution shift is a meaningful and contemporary topic, especially for fairness and online prediction settings.
2. Its attempt to bring online multiobjective theory closer to empirical evaluation is welcome in a literature often dominated by proofs.
3. The paper is well-written and logically structured, with clear definitions, algorithms, and theorems that make the technical content easy to follow.

**Weaknesses:**

1. The main algorithm is only a minor variation of existing methods such as the exponential-weights and Fixed-Share frameworks from prior work (e.g., Lee et al., 2022; Gradu et al., 2023). The theoretical results follow known patterns without introducing new proof techniques or bounds.
2. The claimed “multi-objective” setup is misleading since all objectives depend on the same residual term. It can hardly be generalized to multi-objective setting with competing objectives.
3. The experiments are limited to two small datasets, with mostly qualitative plots and no statistical analysis or ablations on key hyperparameters.
4. The connection between “local adaptivity” and true robustness to arbitrary distribution shift is not theoretically justified.

**Questions:**

1. In line 121 $\ell:[0,1]\times X \times Y \rightarrow [-1,1]$ takes $p_t$ as the first input, while in line 119, it says $p_t\in Y=[a,b]$. Then why it is  [0,1] here? And, in Definition 1, it says $L = \{\ell:[0,1]\times X \times Y \rightarrow [0,1]\}$ why the output is $[0,1]$ rather than $[-1,1]$ now? And is Definition 1 a standard definition for online multi-objective learning?
2.  Is Definition 1 a standard definition for online multi-objective learning? And can you provide justification or references for Defs 2 and 3 as well?
3. How sensitive are results to the interval width and exploration rate?
4. Why the objectives can be simply renormalized to [−1,1] rather than balanced by variance or importance?
5. Can the algorithm handle noncontiguous or abrupt distribution shifts, and how does it compare to other adaptive fairness methods?

---

> ### Author Response · Authors · 2025-11-24
> **Author Response (1/2)**
>
> We thank the reviewer for their thoughtful comments and feedback. We address specific questions and remarks (quoted) below.
>
> > The theoretical results follow known patterns without introducing new proof techniques or bounds.
>
> The focus of our paper is connecting ideas across multi-objective learning and adaptive online learning literature to propose performant methods in practice. We perform extensive empirical comparisons largely missing in the literature until now and present a locally adaptive method that outperforms the previous proposals. Our goal is to serve as an initial step toward bridging the empirical gap in this line of work that is theoretically powerful and promising. As noted by the reviewers, we believe this is an important contribution to this literature.
>
> > The claimed “multi-objective” setup is misleading since all objectives depend on the same residual term. It can hardly be generalized to multi-objective setting with competing objectives.
>
> We respectfully disagree that the “multi-objective” setup is misleading. Our framework is general and can incorporate any set of self-consistent objectives $\ell$ that share a common minimizer. It is true that our framework cannot handle competing objectives. However, the problem we’re studying here captures a wide array of common tasks in the literature (e.g., multiaccuracy, multicalibration, omniprediction, multi-group learning) that are of broad interest. Additionally, even though the objectives are self-consistent, it is still difficult to design algorithms that guarantee strong performance over all of them simultaneously. This has been a problem of wide study in recent literature.
>
> > The experiments are limited to two small datasets, with mostly qualitative plots and no statistical analysis or ablations on key hyperparameters.
>
> We have significantly extended our empirical evaluations. Please refer to the global response (1) for a detailed summary of the experiments we have added to the updated paper. Three particular changes we have made that directly address the above are:
>
> 1. We have added ablations on various hyperparameters including the number of buckets ($m$) (Appendix D.2), interval width $|I|$ (Appendix F.1), $\eta$ (Appendix F.2), and $\gamma$ and $\tau$ (Appendix F.3).
>
> 2. We have added quantitative plots that report the total multiaccuracy over the windows. We include results over all methods with varying window widths $|I|$ (Figure 10), choices of $\eta$ (Figure 12), and choices of the fixed interval width $\tau$  (Figure 13).
>
> 3. We have added additional results on simulated data (Appendix G).
>
> > The connection between “local adaptivity” and true robustness to arbitrary distribution shift is not theoretically justified.
>
>
> Thank you for the question. The way we view distribution shifts here is that the features $x_t$ and labels $y_t$ are drawn from a distribution $Q_t$ that can shift arbitrarily, i.e., $(x_t, y_t) \sim Q_t$ where $Q_t$ is changing. The adversarial setting is a broader setting that captures both this model and many other data distributions.
>
> Theorem 2 in the paper guarantees that under arbitrary distribution shifts, the maximum accumulated value over the objectives in local intervals of fixed width $|I|$ scales with $O(\sqrt{|I| \log (|\mathcal{L}|\cdot|I|)})$. Informally, local adaptivity guarantees that the local multiaccuracy error is small and predictors are robust to rapid changes in the distribution.
>
>
> > In line 121 $\ell: [0,1] \times \mathcal{X} \times \mathcal{Y} \rightarrow [-1, 1]$ takes $p_t$ as the first input, while in line 119, it says $p_t \in \mathcal{Y} = [a,b]$. Then why it is [0,1] here?
>
> This was a typo. Our predictor $p_t$ always takes values in the same domain as $y_t$. We thank the reviewer for pointing out this error.
>
> > In Definition 1, it says $\mathcal{L} = \{\ell: [0,1] \times \mathcal{X} \times \mathcal{Y} \rightarrow [0, 1]\}$ why the output is [0, 1] rather than [-1, 1] now?
>
> You’re correct, that is a typo. Thanks for pointing it out, we have fixed it.
>
> > Is Definition 1 a standard definition for online multi-objective learning? And can you provide justification or references for Defs 2 and 3 as well?
>
> Yes, the learner’s goal in online multi-objective learning has been defined previously in Lee et al., 2022 as minimizing the maximum accumulated value over each of the objectives and captures many common problems of interest.
>
>  As suggested, we have added additional citations for Definitions 2 and 3 in the updated version of the paper.

---

> ### Author Response · Authors · 2025-11-24
> **Author Response (2/2)**
>
> > How sensitive are results to the interval width and exploration rate?
>
> Thank you for the question. We have added empirical evaluations that address this question. Please see Appendix F.1 for ablations on varying interval width $|I|$ and Appendix F.3 for ablations on exploration rate. The results show that our proposed locally adaptivity algorithm consistently outperforms all other adaptive algorithms despite being tuned with a fixed width. In addition, the results are quite robust to both the interval width used to tune the algorithm and the corresponding exploration rate $\gamma$.
>
> > Why the objectives can be simply renormalized to [−1,1] rather than balanced by variance or importance?
>
> The objectives are indeed reweighted by importance in the online algorithm (Algorithm 1, lines 3-5). The probability distribution $q^{(t)}$ over the set of objectives reweights the objectives such that objectives that incurred larger value are assigned greater probability.
>
> > Can the algorithm handle noncontiguous or abrupt distribution shifts, and how does it compare to other adaptive fairness methods?
>
> Yes, the algorithm can handle noncontiguous or abrupt distribution shifts and outperforms other methods. We have added a new section in the appendix (Appendix G) for simulated examples to demonstrate this. We consider different magnitudes of distribution shift, particularly jump discontinuities of various sizes. Our results show that the proposed locally adaptive algorithm is able to adapt to discontinuous and abrupt distribution shifts with better rates than existing methods, and that this difference is substantial when the size of the jump shift is large.
>
>
> We hope our responses address the concerns sufficiently. Please let us know if there is anything more we can do to further address your concerns or provide additional clarity.

---

### Official Review · Reviewer_Jrda · 2025-11-01

**Soundness:** 3
**Presentation:** 3
**Contribution:** 2
**Rating:** 4
**Confidence:** 4

**Summary:**

The paper considers online minimax multiobjective optimization, and proposes an algorithm that offers locally adaptive regret guarantees (which hold not only marginally but also over windows of some size). The paper develops theoretical guarantees for local adaptivity, that combine the insights from Lee et al (2022) with control-inspired adaptive methods of Gradu et al (2023). It then tests the proposed approach on two datasets, a fairness one and an electric one, and finds a substantial empirical advantage of the proposed method over the previous algorithm, i.a. when it is geared towards optimizing for multiaccuracy + regret.

**Strengths:**

The proposed method appears to be an empirically advantageous one compared to prior work, naturally iterating on the original work of Lee et al. The algorithm remains clean and interpretable. Several practical tweaks, such as adding in regret constraints and using the step size heuristic from the adaptive conformal literature, appear to be fruitful at enhancing performance.

Furthermore, some of the empirical findings (i.e. the faster convergence, to multi accuracy, of multiaccuracy-only approaches compared to multicalibration ones) add to the recently accumulated evidence on the theory/empirical fronts that there are interesting separations between multiaccuracy and multicalibration notions.

As another point to put this work into perspective, the multivalidity / multicalibration literature has long been developed with mostly theoretical guarantees in mind, and a variety of initially developed methods have suffered from various performance issues (such as overfitting in the batch case, and slow empirical convergence in the online case), which have only recently received attention from the empirical angle. These recent works have focused on the batch case, primarily, and so the development of an empirically performant online method is a welcome and timely development from that broader viewpoint.

**Weaknesses:**

The proposed method, while a rigorous one and an apparent improvement over the state of the art, appears to be a somewhat modest development on the theory front. I have read the proofs (and found them to be correct), but have not found that many surprising insights. In a way, it is natural with respect to the online multiobjective framework that there are improvements of the adaptive regret style (indeed, these can just be enforced via separate constraints, ignoring the worse theoretical/empirical guarantees that this would obtain); what would have been more exciting for me to see would be strong dynamic regret results: indeed, short of the well-known blackbox adaptive <-> dynamic regret conversions, it’s not quite clear to me how to naturally embed dynamic guarantees into an explicitly minimax-solving framework like the one studied here.

Moreover, I think the experiments could also be expanded on in a fruitful manner, as I was left with some questions on the “practical” front; here are a few points. First, note that multicalibration is separated from multiaccuracy via the tunable parameter m (number of buckets), such that when there are no buckets we get multiaccuracy. So in terms of the multiaccuracy/multicalibration separation, it would be very useful to show some plots where less and less multicalibration constraints are imposed, and seeing how that reflects on the obtained multiaccuracy convergence rates; seeing this at least for small bucket counts (2, 3, 5, etc) would be informative. Second, while the learning rate choice as in Gibbs and Candes is a solid one, carefully ablating against it would be important to see to what extent that choice matters compared to the locally adaptive updates themselves. Third, given the partly theoretical nature of the main contribution, it would be good to see some synthetic experiments besides real-world ones, e.g. exploring the speed of data drift impacting the new and the old algorithms.

**Questions:**

First, please see the empirical questions above. Second, I have not found this in the paper in long/explicit form, but it would be good, exposition-wise, to explain to the readers how local adaptivity can be obtained in the existing original framework via separate constraints, namely, by just instantiating constraints (that come and go every length-of-window many rounds) for each of the currently active windows; explaining and perhaps even showcasing via a quick experiment why that would be less practical.

---

> ### Author Response · Authors · 2025-11-24
> **Author Response**
>
> We thank the reviewer for their thoughtful comments, in particular for recognizing the significance of multiaccuracy approaches and appreciating the timely contributions. We address specific questions and remarks (quoted) below.
>
>
> > what would have been more exciting for me to see would be strong dynamic regret results: indeed, short of the well-known blackbox adaptive <-> dynamic regret conversions, it’s not quite clear to me how to naturally embed dynamic guarantees into an explicitly minimax-solving framework like the one studied here.
>
> We agree that this is a good suggestion and worthy of further investigation. At this stage, our proposed meta-algorithm (Algorithm 1) can be used with any general procedure for learning weights by replacing the hedge algorithm with a locally adaptive method. While we focus on the Fixed Share method in our work, an alternative choice is strongly adaptive learning methods as in Daniely et al. (2015). Prior work [1] has discussed the relationship between strongly adaptive and dynamic regret. However, this was not the focus of our current work.
>
> [1] Zhang, L., Yang, T., Jin, R., & Zhou, Z. (2017). Dynamic Regret of Strongly Adaptive Methods. International Conference on Machine Learning.
>
> > Expanding the experiments
>
> Thank you for your suggestions. We have taken this feedback into account and significantly extended our empirical evaluations: see (1) of our global response. Here, we specifically go over the experiments suggested in your review, which is one component of our broader effort to expand the empirical results.
>
> 1. **Varying $m$:** We evaluate the multiaccuracy error for varying number of buckets $m$ in Appendix D.2. The results show that as the number of buckets decreases, the multiaccuracy error of the multicalibration algorithm goes down, which aligns with the theory.
>
> 2. **Varying $\eta$:** We now run ablations of the locally adaptive algorithm with two other choices of $\eta$ alongside the proposed adaptive $\eta_t$ (eq. 9): (i) the value $\eta = \sqrt{\frac{\log|\mathcal{L}|}{T}}$ which gives the optimal non-adaptive error bound  and (ii) $\eta = \sqrt{\frac{\log(|\mathcal{L}|\cdot 2\tau) + 1}{\tau}}$, which substitutes the adaptive online updates with the fixed interval width $\tau$ (see Appendix F.2 for details). Our results show that adaptive $\eta_t$ consistently dominates, establishing the importance of this choice in achieving local adaptivity separate from the uniform exploration.
>
> 3. **Synthetic experiments:** Synthetic experiments would certainly be valuable. We have added a new section in the appendix (Appendix G) for simulated examples. We compare the algorithms under settings with different magnitudes of the distribution shift. Our results show that the proposed locally adaptive algorithm is able to adapt to discontinuous and abrupt distribution shifts with better rates than existing methods, and that this difference is substantial when the size of the distribution shift is large.
>
> > it would be good, exposition-wise, to explain to the readers how local adaptivity can be obtained in the existing original framework via separate constraints, namely, by just instantiating constraints (that come and go every length-of-window many rounds) for each of the currently active windows; explaining and perhaps even showcasing via a quick experiment why that would be less practical.)
>
> We appreciate the suggestion. This is exactly the method proposed in prior work that we compare with. We have edited our introduction to clarify this, as well as added a new subsection titled ‘**A closer look at adaptive algorithms in literature**’ (Section 2.3) in the revised version of the paper.
>
> We compare against this method in all our experiments. In addition, we have included **new experiments in Appendix E** where we compare the proposed approach with the multiaccuracy algorithm that uses the adaptive method suggested in prior work. Our results show that our proposed method consistently outperforms this approach in a variety of settings (see in particular Figures 4-5 and 8-10).
>
> We hope our responses address the concerns sufficiently. Please let us know if there is anything more we can do to further address your concerns or provide additional clarity.

---

### Official Review · Reviewer_6q46 · 2025-11-01

**Soundness:** 2
**Presentation:** 2
**Contribution:** 2
**Rating:** 2
**Confidence:** 4

**Summary:**

The paper presents adaptive algorithms for multi-objective learning to guarantee controlled error across a set of objectives over local time intervals in sequential problems. The paper also does experimental studies on electric load forecasting and compas dataset, and compares with prior work from Lee et al. (2022) and other strong adaptive variants proposed in the literature.

**Strengths:**

The paper has some merit as it attempts to initiate empirical studies on this line of work. However, I do have some concerns as noted below.

**Weaknesses:**

For me it is not clear what is the main contribution of the paper. The introduction says: the majority of algorithms proposed in the literature are not adaptive, but then then around lines 247 mentions strong adaptivity has been previously studied before, and the related extensions have been studied for multi-calibration. In experimental section, comparison against strong adaptivity is done, but I'd expect some more insightful resolution on why that does not work in practice as the experiment section claim. Maybe this needs to be studied a bit more.

Additionally, I'm bit confused about the setting of the paper. I assume the related work in this line assumes an adversarial form of nature, i.e. there is no distribution, and the outcome is determined by an adversarial nature. In that sense, what does it mean to have distribution shift. The example given at the beginning of Section 2.2, I believe the online algorithms proposed in the literature would be able to cope with it, although there'd be some delay for $t > T/2$. I get the motivation of the work that vanishing error (multi-accuracy or multi-calibration error) in the limit does not say much about error in local intervals, which makes up for a good motivation for the paper, but distribution shift argument is something I'd appreciate clarification about.

My reading of the paper says that the paper aims to provide experimental evidence or is intended as a practical study of multi-objective algorithms in the online setting with adaptiveness as a major focus, however the current setup seems lacking as I could not learn why prior algorithms (multi-calibration and strong adaptive variant of it) does not work, and what does the algorithm proposed in the paper contributes that gives better performance. I feel like this requires more study.

My preliminary impression of the paper is that while the paper has good potential as a rather insightful contribution in this literature, the paper requires more thorough work (experimental and maybe theoretical to compare against prior work) to add value to the literature. And I hope authors will consider this.

**Questions:**

See above

---

> ### Author Response · Authors · 2025-11-24
> **Author Response**
>
> We thank the reviewer for their thoughtful comments and feedback. We address specific questions and remarks (quoted) below.
>
>
> > Clarifying contributions; why prior algorithms and adaptive versions do not give better performance
>
> We agree this requires more clarification and explanation. Firstly, we have made significant edits to the abstract and introduction to clarify our contributions. Second, we have added a new subsection titled ‘**A closer look at adaptive algorithms in literature**’ (Section 2.3) in the revised version of the paper.
>
> The only prior adaptive method for online multi-objective learning we are aware of is from Lee et al., 2022. Their method lends adaptivity to existing multi-objective algorithms by including additional objectives for all subintervals. We present an alternative approach that directly modifies the multi-objective learning algorithm by replacing one part of the scheme with an adaptive online learning method. For a fixed interval width $|I|$, our proposed method guarantees that the local multiobjective error is at most $O(\sqrt{|I| \log (|\mathcal{L}|\cdot|I|)})$, while for the method proposed in Lee et al. the bound is $O(\sqrt{T(\log |\mathcal{L}| + 2 \log T)})$. The additional objectives introduced by including all contiguous intervals make the multi-objective problem more complex and the difference between the bounds is borne out in practice in the experiments. This is confirmed by our experimental results in which we find that our approach outperforms the method of Lee et al., 2022 across a variety of settings (see in particular Figures 4-5 and 8-10). Additional details outlining the difference between the two approaches are given in the updated abstract and introduction and in Section 2.3.
>
>
> > Clarification about distribution shift
>
> Thank you for the question. The way we view distribution shifts here is that the features $x_t$ and labels $y_t$ are drawn from a distribution $Q_t$ that can shift arbitrarily, i.e., $(x_t, y_t) \sim Q_t$ where $Q_t$ is changing. The adversarial setting is a useful mathematical tool that captures both this setting as well as many others.
>
> You’re correct—the non-adaptive algorithms may be able to cope with some delay in this simple illustrative example. However, this delay is not desirable; moreover, when the distribution shifts rapidly and the optimal predictor changes often, non-adaptive algorithms will not be able to cope with the shifts. Our results on real datasets with distribution shifts show that non-adaptive algorithms have significantly slower rates in practice.
>
> We have now also added simulated examples in Appendix G where we consider different magnitudes of shift in $Q_t$. Our results show that the proposed locally adaptive algorithm is able to adapt to discontinuous and abrupt distribution shifts with better rates than existing methods, and that this difference is substantial when the size of the distribution shift is large.
>
> Thank you again for your feedback. Please let us know if there is anything more we can do to further address your concerns or provide additional clarity.

---

### Author Response · Authors · 2025-11-24
**Global Author Response**

We are grateful to the four reviewers who took interest in the paper and suggested concrete ways to improve it. Thank you!

We have made major additions and uploaded a revised draft. We have highlighted text corresponding to specific changes asked for by the reviewers in blue. Some additional changes have been made in other sections to clarify the exposition.

The reviewers found the work timely and the contribution practically important in a literature that has largely been theoretical until now.

There were two main concerns: lack of ablations and limited discussion of prior adaptive algorithms in literature. We have taken the feedback quite seriously and incorporated the following extensive revisions in the new version:

1. **Major experimental additions**

- We have now added **ablations on all hyperparameters**. See Appendix D.2 for ablations on the multicalibration buckets $m$, Appendix F.1 for ablations on the interval width $|I|$, Appendix F.2 for experiments with varying choice of $\eta$, and Appendix F.3 for analysis on sensitivity to $\gamma$. Results show that the proposed locally adaptive algorithm consistently outperforms other adaptive methods and is robust to hyperparameter choices.

- We added a **new section for simulated examples** (Appendix G) where we consider different magnitudes of distribution shift and compare the local adaptivity of all methods. Results show that the proposed locally adaptive algorithm is able to adapt to discontinuous and abrupt distribution shifts with better rates than existing methods.

- We have included **quantitative plots** that report the total multiaccuracy error over the windows. We include results over all methods with varying interval widths (Figures 9, 10) , choices of $\eta$ (Figure 12), and $\tau$ (Figure 13).


2. **Discussion of adaptive methods in literature**

- We have updated the introduction and abstract to make our contributions clearer and explain the adaptive method proposed in prior work.

- We have added a **new subsection titled ‘A closer look at adaptive algorithms in literature’** (Section 2.3) in the revised version of the paper that discusses some examples of adaptive methods in the online learning literature.

- We have now also added empirical comparisons with two additional adaptive baselines: (i) MC (locally adaptive) algorithm, which uses our proposed locally adaptive approach in Algorithm 2 for multicalibration objectives, and (ii) MA+reg (adaptive) algorithm, which uses additional objectives for intervals as proposed in prior work (Lee et al., 2022). Our results indicate that our proposed method consistently outperforms all existing baselines across a variety of settings.


We thank the reviewers again for their helpful feedback. We hope this addresses the main concerns, and we respond to the individual concerns in separate comments.

---

### Author Response · Authors · 2025-12-03
**Summary note**

We thank the AC for leading the review process and the reviewers for their concrete feedback and questions. We believe we have addressed the reviewers’ concerns and incorporated all feedback through extensive revisions and new experiments.

Reviewers appreciated our contribution as timely and practically important in a growing line of work that has been largely theoretical to date. Below, we summarize how our rebuttal and revisions address the main points raised by the reviewers.

1. **Discussion of adaptive algorithms in prior work.** Reviewers 6q46 and Jrda requested additional explanation of the adaptive method from prior work in Lee et al., 2022 and clarification on why it does not perform better. We added a new subsection titled ‘A closer look at adaptive algorithms in literature’ (Section 2.3) and explained that the difference between the local multiobjective error bounds of the methods is borne out in practice in the experiments (see the next point). In addition, we compare against this method in all our experiments and show that our proposed approach consistently outperforms the previous proposal across a variety of settings (see in particular Figures 4-5 and 8-10).

2. **Clarifying our contributions.** Reviewer 6q46 asked for clarification on the main contribution of the work. We have updated the introduction and abstract to make our contributions clearer. Previous work (Lee et al., 2022) lends adaptivity to existing multi-objective algorithms by including additional objectives for all subintervals. We present an alternative approach that directly modifies the multi-objective learning algorithm by replacing one part of the scheme with an adaptive online learning method. For a fixed interval width $|I|$, our proposed method guarantees that the local multiobjective error is at most $O(\sqrt{|I| \log (|\mathcal{L}|\cdot|I|)})$, while for the method proposed in Lee et al. the bound is $O(\sqrt{T(\log |\mathcal{L}| + 2 \log T)})$.

3. **Clarification about distribution shift and whether the proposed algorithm can handle noncontiguous and abrupt distribution shifts.** Reviewers 6q46 and iN2x asked for clarification on how we view distribution shifts in our work, and on the connection between local adaptivity and robustness to distribution shift. We respond to these clarifying questions in individual comments to the reviewers. In addition, reviewers Jrda and iN2x suggested investigating whether the proposed algorithm can handle abrupt distribution shifts and how its performance compares with other adaptive methods. We added a new section (Appendix G) for simulated examples to demonstrate this. We consider different magnitudes of distribution shift, particularly jump discontinuities of various sizes, and show that the proposed locally adaptive algorithm is able to adapt to discontinuous and abrupt distribution shifts with better rates than existing methods.

4. **Missing ablations and quantitative plots.** Reviewers Jrda and iN2x asked for ablations on hyperparameters and quantitative plots. In response, we included the requested ablations and quantitative analyses, as well as additional empirical evaluations. Please see global author response (1) for a detailed summary of the experiments we have added to the updated paper.

5. **Clarification on adaptive $\eta_t$.** During the initial discussion period, we sought to address Reviewer L6nz’s misunderstanding about how the adaptive $\eta_t$ is set. As we state in our response to the reviewer, we do not require knowledge of the specific interval in our algorithm. We use the interval length $|I| = \tau$ to set an adaptive value of $\eta_t$ that tracks changes in the moving average of the expected squared objective over the most recent $ \tau$ time steps.

---

### Meta-Review · Area_Chair_jKTe · 2026-01-09

**Summary:**

This paper addresses the problem of multi-objective learning, in which a single model simultaneously satisfies multiple learning goals (loss functions). The Authors consider an online setting where the data distribution can change arbitrarily over time. The proposed algorithm provides locally adaptive regret guarantees that hold for contiguous subintervals. The proven theoretical guarantees concern local adaptivity. The paper also includes an extensive empirical study on two datasets.

The paper is appreciated by the Reviewers as a timely and practically important contribution, however, all initial scores are negative. The main issues raised concern an unclear contribution and problem setup, insufficient discussion of related work, limited theoretical novelty, and an experimental evaluation lacking adequate ablation studies. One Reviewer also noted that referring to the problem as multi-objective learning may be misleading. I share this concern, as traditional multi-objective optimization typically focuses on finding trade-offs by selecting points on a Pareto frontier. Nevertheless, this terminology is used consistently within this particular area of the machine learning literature.

The Authors prepared an extensive rebuttal containing many interesting points. Unfortunately, this paper may be a victim of the OpenReview data leak and the associated changes to the reviewing process. It is very likely that the Reviewers would increase their initial scores during the discussion, however, it is less certain that the scores would cross the acceptance threshold, given the highly competitive nature of ICLR. The paper can still benefit substantially from an additional revision in which the main messages of the Authors should be articulated more clearly.

**Reviewer Concerns:**

Some concerns, such as limited theoretical novelty, would likely remain, but many others have been adequately addressed by the authors.

**Reviewer Scores:**

- 6q46 would likely increase the score to 4 or 5
- Jrda might raise the score to 5 or 6
- iN2x might raise the score to 5 to 6
- L6nz might raise the score to 5 to 6

---

### Decision · Program_Chairs · 2026-01-26

Reject